# DIS-CO: Discovering Copyrighted Content in VLMs Training Data

**André V. Duarte** [1][2]  **Xuandong Zhao** [3]  **Arlindo L. Oliveira** [2]  **Lei Li** [1]

## Abstract

*How can we verify whether copyrighted content was used to train a large vision-language model (VLM) without direct access to its training data?* Motivated by the hypothesis that a VLM is able to recognize images from its training corpus, we propose DIS-CO, a novel approach to infer the inclusion of copyrighted content during the model's development. By repeatedly querying a VLM with specific frames from targeted copyrighted material, DIS-CO extracts the content's identity through free-form text completions. To assess its effectiveness, we introduce Movie-Tection, a benchmark comprising 14,000 frames paired with detailed captions, drawn from films released both before and after a model's training cutoff. Our results show that DIS-CO significantly improves detection performance, nearly doubling the average AUC of the best prior method on models with logits available. Our findings also highlight a broader concern: all tested models appear to have been exposed to some extent to copyrighted content. Our code and data are available at https://github.com/avduarte333/DIS-CO

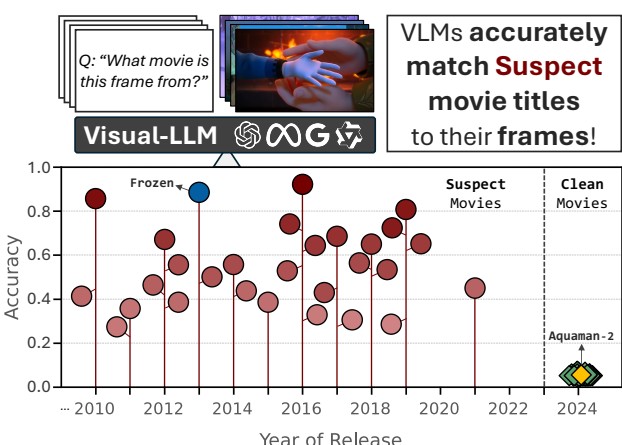

*Figure 1.* Our DIS-CO reveals that VLMs successfully map frames of suspect movies to their titles, even when the frames are highly challenging. For example, the GPT-4o model accurately identifies the movie "Frozen" (blue circle), despite the varying complexity of its frames. In contrast, for newly released films, the models are unable to perform similar frame-to-title predictions.

## 1. Introduction

The rapid evolution of large-scale models has driven a paradigm shift toward multimodality, with recent large vision-language models (VLMs) gaining prominence for their ability to process both visual and textual information (Alayrac et al., 2022; Liu et al., 2023; OpenAI, 2023; Wang et al., 2024). While these models showcase remarkable performance across a variety of tasks, their reliance on vast, diverse datasets introduces challenges in ensuring compliance with ethical and legal standards. Without strict safeguards during the data collection step, proprietary content could be incorporated into the models' knowledge, opening the door to intellectual property infringements and potential legal conflicts (Carlini et al., 2022a; Nasr et al., 2023; Duan et al., 2024). In fact, in the United States alone, more than 24 copyright lawsuits were filed against the AI industry since 2023 (Knibbs, 2024), reflecting growing concerns about the use of protected material in training (Kadrey, 2023; Daily News, 2024).

Discovering training data is, therefore, essential for effectively addressing the ethical and legal challenges of model training. However, the lack of transparency in data collection (often justified by competitive concerns) makes it especially difficult to trace the inclusion of specific content.

To tackle these challenges, Membership Inference Attacks (MIAs) serve as a tool to identify whether specific data samples were part of a model's training set. While MIA techniques are well-studied for text-based models, their adaptation to multimodal settings, particularly VLMs, remains less explored - a gap that our work aims to address.

[1]Carnegie Mellon University [2]INESC-ID / Instituto Superior Técnico, ULisboa [3]UC Berkeley. Correspondence to: André V. Duarte <andre.v.duarte@tecnico.ulisboa.pt>, Xuandong Zhao <xuandongzhao@berkeley.edu>, Arlindo L. Oliveira <arlindo.oliveira@tecnico.ulisboa.pt>, Lei Li <leili@cs.cmu.edu>.

*Proceedings of the $42^{nd}$ International Conference on Machine Learning*, Vancouver, Canada. PMLR 267, 2025. Copyright 2025 by the author(s).

Recent advancements have been made in this field, but certain challenges and limitations continue to exist. The $\mathrm{MaxR\acute{e}nyi\text{-}K\%}$ method (Li et al., 2024b) is based on the intuition that data encountered during training leads to greater model confidence when generating outputs, reflected by a smaller Rényi entropy in the next-token probability distribution for image or text tokens. While this method proved effective in settings with unrestricted access to output logits, its applicability is limited in the context of black-box models like Gemini (Reid et al., 2024), which, at most, allow for inspection of the top-5 logits for each predicted token. The work of Pinto et al. (2024) addresses the challenge of discovering training data in a fully black-box fashion by introducing a method tailored to document-based VQA tasks. By removing key textual content from input images, the authors demonstrate the models' ability to recall memorized training data, including sensitive information. This approach is, nonetheless, limited to VQA datasets, which constitute only a small fraction of the diverse data types used to train VLMs.

In this paper we propose DIS-CO, a novel method for detecting models' training data that overcomes the limitations of previous approaches while being applicable to both white- and black-box VLMs. The core idea of DIS-CO is to prompt models to map a set of images from a target media document to its corresponding identity (e.g., movie titles) in an unconstrained, free-form text-generation manner, enabling them to produce answers freely instead of selecting from predefined options. As a result, under the null hypothesis: *the target content was not in the training dataset*, the model is much less likely to correctly identify the data, reducing false positives. Consequently, correct outputs become a more reliable indication that the target content was part of the training dataset (Zhang et al., 2024a).

This idea is illustrated in Figure 1, which shows that models solve the task correctly far more often for content that was very likely included in their training data compared to content that was definitely excluded, such as movies released after the model's cutoff knowledge date. For instance, DIS-CO maps frames from the movie *Frozen* to the correct title in nearly 90% of the test cases, while a movie like *Aquaman and the Lost Kingdom* is correctly identified in less than 2% of the time.

We conduct experiments on two benchmarks, MovieTection (our newly introduced dataset) and VL-MIA/Flickr (Li et al., 2024b). MovieTection contains 14,000 diverse movie frames paired with descriptive captions, split chronologically based on films released before/after the models' training cutoff (October 2023). VL-MIA/Flickr, derived from COCO (Lin et al., 2014) (member data) and recent Flickr images (non-member data), serves as a proof-of-validity dataset for DIS-CO.

Our main contributions are as follows:

- We introduce DIS-CO, an innovative method applicable to both white-box and black-box VLMs, designed to detect whether copyrighted content was included in training. To the best of our knowledge, this is the first work to detect copyrighted movies in the context of VLMs.

- We introduce MovieTection, a new benchmark designed for detecting training data of VLMs. The dataset includes content from 100 movie titles, unfolded into 14,000 frames, with a mix of easily identifiable and challenging examples to test model capabilities comprehensively.

- Experiments show that DIS-CO effectively detects copyrighted movies across the six tested model families, with GPT-4o achieving an average accuracy of 34% on the "hard to guess" frames of the suspect movies.

- We show that fine-tuning a model to avoid disclosing memorized content for a particular movie is an effective defense strategy, capable of generalization to other titles.

## 2. Preliminary and Related Work

Membership Inference Attacks (MIAs) are designed to determine whether a specific data instance was included in the training set of a machine learning model (Shokri et al., 2017; Hu et al., 2022). This area of research has seen growing interest with the increasing use of LLMs, which are known to memorize and occasionally reproduce training data (Nasr et al., 2023; Carlini et al., 2022b; Hans et al., 2024).

Classical MIAs are typically divided into two main approaches: reference-based and reference-free. Reference-based methods involve training a set of "shadow models" to replicate the target model's behavior (Carlini et al., 2022a; Long et al., 2018; Mireshghallah et al., 2022; Watson et al., 2022). In contrast, reference-free methods rely on calculating specific metrics, such as the perplexity of a sentence, to identify patterns indicative of training set membership (Yeom et al., 2018; Salem et al., 2018; Carlini et al., 2020; Song & Mittal, 2021). Among these, the $\mathrm{Min\text{-}K\%\text{-}Prob}$ method stands out as a more refined approach. It hypothesizes that the average log-likelihood of the top-k% least probable tokens in an example is higher if the example was part of the training data compared to if it was not (Shi et al., 2023). Building on this foundation, recent extensions such as $\mathrm{Min\text{-}K\%\text{++}}$ (Zhang et al., 2024b) and DC-PDD (Zhang et al., 2024c) have introduced further improvements. However, a key limitation of most reference-free methods is their dependence on access to token probability distributions, which restricts their interoperability with black-box models such as Gemini (Reid et al., 2024).

With recent research shifting focus from text-only models to multi-modal architectures, the task of detecting training data and evaluating model memorization has begun to emerge

in this domain as well (Kokhlikyan et al., 2024; Jayaraman et al., 2024; Pinto et al., 2024). Building on techniques originally developed for text-only models, Li et al. (2024b) propose a novel image-based MIA pipeline that adapts methods like $\mathrm{Min\text{-}K\%\text{-}Prob}$ (Shi et al., 2023) to VLMs. The work presents the $\mathrm{MaxR\acute{e}nyi\text{-}K\%}$ metric, which enables image membership inference by analyzing the output logits corresponding to the model's image-specific slice.

Detecting training data is especially significant when it involves copyrighted content, as the reproduction of such material by large models raises legal and ethical concerns (Li et al., 2024a; Meeus et al., 2024a). In the light of counterfactual memorization studies, the methods proposed by Duarte et al. (2024) and Golchin & Surdeanu (2024) perform membership inference through a multiple-choice question-answering (MCQA) setting. These approaches demonstrate solid results and have the advantage of being applicable to both white-box and black-box models, as they do not depend on access to token probabilities. However, it is known that multiple-choice scenarios may induce a selection bias (Zheng et al., 2024a) on the models, which introduces some uncertainty about whether chance played a role in the results. In contrast, Karamolegkou et al. (2023) adopt a prompting approach with free-form text generation, aiming to elicit verbatim reproduction of copyrighted material. This provides stronger evidence of memorization because, in an unconstrained, free-form setting, the model is much less likely to produce correct outputs by chance. While this method may fail to detect cases where models are trained on copyrighted data without memorizing it (Meeus et al., 2024b), it also avoids the issues raised by Das et al. (2024) and Maini et al. (2024), who warn that many membership inference methods risk overstating results by exploiting data distribution shifts, such as temporal patterns, rather than identifying genuine memorization. For these reasons, we also focus on free-form text generation in this work, as it provides a more robust and unbiased indication of whether the target content is part of the model's training data.

## 3. Benchmark: MovieTection

Our proposed benchmark, MovieTection, distinguishes member and non-member data based on a clearly defined temporal constraint. Movies released in 2024 or later are considered non-member data, as they fall outside the knowledge cutoff dates of all tested models. Movies from January to September 2023 are excluded due to uncertainty regarding models' exposure to content from that period. For instance, Qwen2-VL (Wang et al., 2024) reports a knowledge cutoff in June 2023. Movies released on or before 2022 are treated as potential member data, as they are more likely to have been included in the training datasets of such models.

MovieTection currently comprises frames from 100 movies,

with plans for future expansion. The selection of movies incorporated into the benchmark is guided by their status as box office hits, based on the assumption that highly popular movies, due to their widespread availability, are more likely to appear in training datasets. For the suspect data, we primarily select titles randomly from the Box Office Mojo's[1] list of the all-time highest-grossing films, with some exceptions to accommodate specific experiments, such as analyzing the impact of IMDb[2] ratings (Section 5). For the clean data, we sample most titles from the Box Office Mojo's list of the highest-grossing films of 2024.

For each movie, we extract frames categorized into two types: main frames and neutral frames. This categorization is designed to introduce varying levels of difficulty for assessing a model's knowledge about a movie. Main frames typically feature key characters to the movie's plot. These frames are intended to be easily recognizable by viewers familiar with the movie. In contrast, neutral frames focus on ordinary visuals, such as landscapes, objects, or minor characters, that are not strongly tied to the movie's narrative. Neutral frames are designed to present a significantly higher challenge, as they rely on subtle contextual cues that are almost impossible to associate with the correct title without prior knowledge of the movie. Figure 2 illustrates the two frame types.

Each extracted frame is accompanied by a detailed caption, generated using the 7B version of the Qwen2-VL model. The prompt used for caption generation, along with an example, is provided in Appendix B. In total, 140 frames are extracted per movie, comprising 100 main frames and 40 neutral ones.

## 4. DIS-CO

Our proposed method, DIS-CO, determines whether examples are memorized by evaluating the model's performance on a question-answering task with free-form text responses. The task we propose involves models performing accurate identification of the content's identity, which, in the case of copyrighted movie identification, corresponds to correctly identifying the movie title. We operate under the premise that models map a frame to the appropriate title far more reliably when that movie is included in their training data compared to when it is not.

In order to perform this mapping, we argue that eliciting free-form (FF) completions is preferable to a multiple-choice question-answering (MCQA) format because it significantly reduces the influence of "luck" associated with guessing. More concretely, consider the probability of a model selecting the correct answer purely by chance in an MCQA

---

[1] https://www.boxofficemojo.com/
[2] https://www.imdb.com/

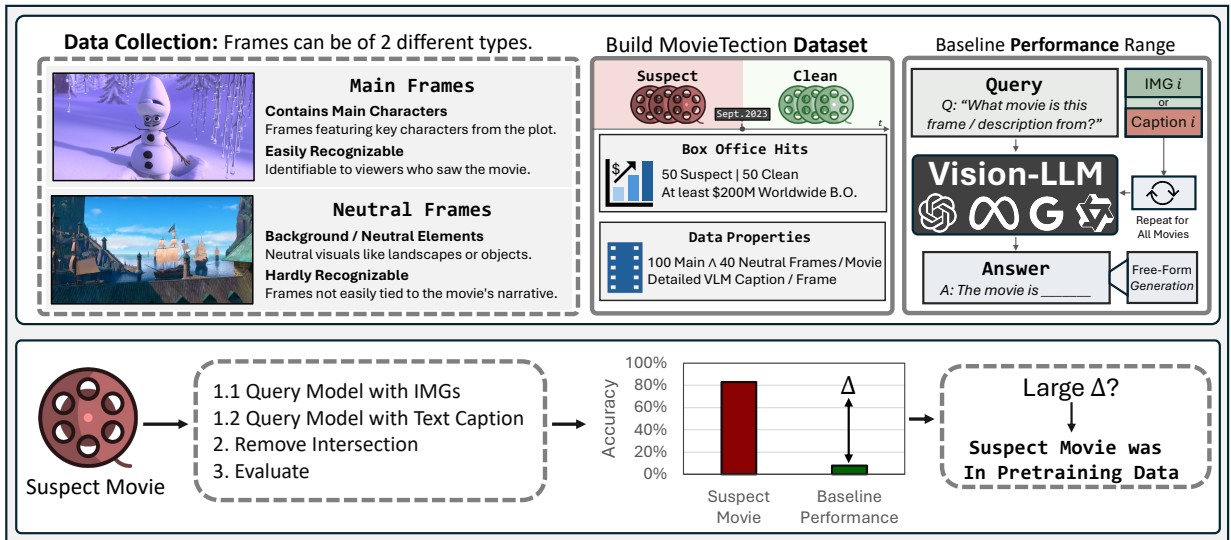

*Figure 2.* The pipeline begins with the construction of the MovieTection benchmark, where we categorize movie frames into main and neutral types to introduce varying levels of difficulty. Models are then queried with image frames and their corresponding captions, producing predictions for both modalities. Predictions from image frames that overlap with caption-based predictions are excluded, isolating cases where image-based memorization is inferred. Performance on the suspect movie is compared against the expected baseline performance, with discrepancies indicating potential training exposure.

scenario with $k$ answer choices. The chance-level accuracy is $P_{\text{MCQA}} = 1/k$, which for the standard case of $k = 4$, results in a baseline accuracy of 25%. This means that even models without any genuine knowledge or memorization of the content can appear to perform well, simply due to the structure of the task.

In contrast, with free-form completions, the models must generate the correct answer from a much larger output space, which, for tasks such as ours, can include thousands of unique possible responses. The probability of producing the correct answer by chance in this unconstrained setting is $P_{FF} \approx 1/|\Omega|$, which, for $|\Omega| = 10{,}000$, corresponds to a baseline accuracy of just 0.01%. Even when accounting for realistic factors such as model bias toward more popular answers, suppose a particular title is 100 times more likely to be generated than an average candidate, the probability $P'_{FF} = 100/10{,}000 = 1\%$ still remains dramatically lower than the multiple-choice baseline. As a result, we have a substantially reduced risk of false positives, and correct predictions are much more likely to reflect genuine memorization.

The overall pipeline of DIS-CO is illustrated in Figure 2. After constructing the MovieTection dataset, we first query the models with clean data to establish a baseline for its expected performance on this set. While one might expect the models to fail completely on all these examples, given that these movies were unreleased at the time of the training cutoff, this is not always the case. Some of the movies in this set, though unreleased, were already announced and

acknowledged by the models, leading to correct predictions for certain examples (See Table 4 - Section 6.3 and Appendix C). Capturing this baseline performance is crucial to avoid incorrectly classifying a movie as part of the training data simply because some frames were accurately identified.

Another important factor to consider is the time effect. In general terms, the older a movie is, the greater the likelihood that a model has residual knowledge about it. This knowledge can come from publicly available online content, such as movie posters, trailers, forum discussions, or datasets like OpenSubtitles (Lison & Tiedemann, 2016), which typically do not raise copyright infringement concerns. To estimate this baseline knowledge accumulated over time, we query the models using the detailed captions from the older movies, as making correct predictions based solely on textual descriptions is unlikely to be problematic.

Finally, to determine whether a specific movie was likely included in the model's training data, we query the model separately with image frames and then with their corresponding caption information from the suspect movie. After both queries are completed, we compare the predictions from these two inputs. If there is an overlap of correct predictions between the frame-based and caption-based queries, we disregard those results, as they suggest the model did not had to rely on the image content to make accurate predictions. By examining the remaining correct predictions, which rely solely on image content, we infer whether the model is utilizing memorized visual information. Ideally, after removing the intersection, the performance of the suspect

movie should fall within this range defined by the baseline performance on the recent movies and the clean baseline performance on older movies with accumulated knowledge over time. However, if performance remains significantly higher than this range, even after removing the intersection, it suggests that the model relied on memorized visual information specific to the movie frames, indicating the movie was likely included in its training data.

The specific prompts used for evaluating models on the MovieTection benchmark are provided in Appendix D.

### 4.1. Upper-Bound Estimation of Memorization

While our proposed approach of removing the intersection between frame-based and caption-based correct predictions provides a more precise set of potentially memorized movies, we cannot rule out that those frames are not part of the training data. Consequently, we also consider an upper-bound estimation of memorization, where all correctly identified frames, regardless of their intersection with captions, contribute to the possible classification of the movie as part of the training data. For clarity, throughout the remainder of this paper, we use two notations: $\lfloor \text{DIS-CO} \rfloor$ represents the smaller set obtained after removing the intersection, while DIS-CO denotes the upper-bound estimation, including all correctly identified frames.

### 4.2. Mitigating the Disclosure of Memorization

While training on copyrighted data may at times be unavoidable, the associated risks can be mitigated by ensuring the model does not disclose memorization. For a movie likely included in the training data, we propose fine-tuning the model on a subset of its frames while replacing the movie label with a neutral designation such as 'Copyrighted Content'. More details on the fine-tuning are in Appendix E.

## 5. Experiments

We assess the effectiveness of DIS-CO through a range of different experiments, which are guided by the following questions:

- **Is DIS-CO suitable for different visual content types?** While our primary focus is on movie frames, VLMs are exposed to a broader range of data during training. As a proof-of-concept, we test whether DIS-CO can detect memorization in other domains, such as comic books and photography.

- **Are factors like movie popularity or quality good proxies for memorization?** To test whether popularity (e.g., box-office revenue) or quality (e.g., IMDb ratings) are proxies for memorization, we collect movies where one factor varies while the other is controlled. For instance,

in the box-office experiment, movies with similar IMDb ratings are chosen to isolate the impact of popularity.

- **Does a longer context reveal more memorization?** As LLMs often perform better with more context in their queries, we hypothesize that VLMs behave similarly. Using the MovieTection dataset, we examine the effect of varying the number of frames in the prompt ($N \in [1, 4]$).

- **How susceptible are models to memorization when exposed to new data?** We investigate the model's ability to memorize new content by fine-tuning it on a movie guaranteed to be outside the training data.

- **How to prevent a model to disclose memorization?** Similarly to the previous experiment, we fine-tune the model (this time on a suspect movie), with a modified labeling objective. This experiment investigates whether this defense mechanism can mitigate memorization disclosure for the suspect movie and whether its effects generalize to other movies.

- **To what extent does generalization influence the model's performance?** Humans are capable of generalizing from partial information, often identifying movies they haven't fully seen by relying on related content such as posters or trailers. To assess how closely models align with humans on this movie detection task, we compare the performance of the models with that of 10 human participants who were selected to identify 200 images from MovieTection.

### 5.1. Experiment Setup

To evaluate DIS-CO, we follow the procedure outlined as follows. Let the "Suspect" group be represented as $S = \{s_1, s_2, \ldots, s_{N_S}\}$ and the "Clean" group as $C = \{c_1, c_2, \ldots, c_{N_C}\}$, where $N_S$ and $N_C$ denote the number of movies in each group, respectively. For each movie, we calculate its accuracy: $A(s_i)$ for $s_i \in S$ and $A(c_j)$ for $c_j \in C$. The accuracy is calculated as the proportion of predictions aligning with the expected outcomes. By default, a weighted average is then applied to account for the unequal proportions of main and neutral frames and the total value is reported. Nonetheless, some results for main and neutral frames are reported individually to provide further insights on the performance across frame types.

We then perform a random sampling process with replacement, repeated 10 times. In each iteration, $M$ elements are sampled from each group, where $M$ corresponds to $N_S$ or $N_C$, depending on the group being sampled. For each iteration, a threshold $\theta$ is optimized to achieve maximum separation between the two groups, and the Area Under the Curve (AUC) is computed.

To complete the analysis, we calculate the mean and standard deviation of the AUC or the average accuracy for the

"Suspect" and "Clean" groups over these iterations. Detection is consistently conducted at the movie level, rather than on individual frames.

### 5.1.1. BENCHMARKS AND BASELINES

We begin by evaluating DIS-CO on two proof-of-concept datasets. The first is a custom-assembled collection of approximately 1,000 comic book pages, sampled from widely recognized series such as *Tintin*, *Asterix*, and *Lucky Luke*. The second dataset, VL-MIA/Flickr, was introduced by Li et al. (2024b) and comprises 600 images divided evenly into "member" and "non-member" categories. Member images are sourced from a subset of COCO (Lin et al., 2014), while non-member images are drawn from recent content posted on Flickr. In both cases, the data is carefully aligned with the knowledge cutoff dates of the models under evaluation.

For the fine-tuning experiments, we use two movies: *IF* (2024) and *Moana* (2016), which have nearly identical durations (1h48min and 1h47min, respectively), allowing us to sample frames at an equal rate, resulting in 6000 frames per movie. The remaining experiments utilize the MovieTection dataset, as detailed in Section 3.

For the main results, we evaluate DIS-CO against three baselines: (i) Captions, (ii) MCQA, and (iii) Rényi ($\alpha = 0.5$). The Captions baseline involves prompting the models using only the textual information available in MovieTection. MCQA presents the models with four possible answers per query, designed to be slightly challenging by including similar movies as distractors (e.g., animated movies are paired with other animated ones). The Rényi baseline applies the Max-K% method ($\alpha = 0.5$) proposed by Li et al. (2024b). We report the results for the value of K that achieves the best detection performance.

## 6. Results

### 6.1. Proof-of-Concept

#### 6.1.1. COMIC-BOOKS

VLMs are designed to handle multimodal content, but their exposure during training is not limited to purely textual or purely visual data; instead, their training corpora often comprises hybrid sources that blend language and images.

Comic books are a natural example of hybrid content, as each page contains both visual illustrations and textual elements. This makes them an ideal setting for our initial proof-of-concept experiment, which focuses on evaluating the applicability of DIS-CO across different types of media.

Table 1 confirms our intuition that models can successfully map comic book pages to their correct titles. For instance, GPT-4o achieves 0.728 accuracy in the MCQA setting and

*Table 1.* Accuracy on the Comics Dataset.

| Method | GPT-4o | Gemini-1.5 Pro | Qwen2-VL 72B |
|--------|--------|----------------|--------------|
| MCQA | $0.728_{0.05}$ | $0.647_{0.04}$ | $0.689_{0.06}$ |
| DIS-CO | $0.538_{0.05}$ | $0.436_{0.07}$ | $0.263_{0.07}$ |

0.538 with DIS-CO's free-form completions: both strong results for this task. While MCQA consistently yields higher absolute accuracy across all models, it is important to interpret this performance gap with caution: the inherent 25% chance-level accuracy in multiple-choice settings can partially inflate MCQA's results. Taking this into account, we believe that both approaches are effectively on par in terms of practical memorization detection. In particular, DIS-CO offers a more reliable indication of genuine content knowledge, as it minimizes the impact of random guessing.

### 6.1.2. VL-MIA/FLICKR

As introduced in Section 5.1.1, VL-MIA/Flickr is an MIA dataset where the 'suspect' images are sourced from COCO (Lin et al., 2014). This proof-of-concept is essential for two key reasons. First, it complements the previous comic-book experiment by further testing DIS-CO's capacity to detect memorization across different domains. Second, and perhaps more importantly, COCO provides a unique validation opportunity: while the inclusion of copyrighted movies or comic books in VLM training data remains uncertain, the presence of COCO images is well-documented in the training data of various models (Radford et al., 2021; Liu et al., 2023), making it an ideal benchmark for DIS-CO. Demonstrating DIS-CO's ability to detect COCO data supports its effectiveness and underscores its potential applicability to similar scenarios.

*Table 2.* Accuracy on the suspect split of VL-MIA/Flickr.

| Method | GPT-4o | Gemini-1.5 Pro | Qwen2-VL 72B |
|--------|--------|----------------|--------------|
| MCQA | $0.020_{0.01}$ | $0.250_{0.00}$ | $0.483_{0.00}$ |
| DIS-CO | $0.413_{0.01}$ | $0.243_{0.00}$ | $0.183_{0.00}$ |

As shown in Table 2, DIS-CO enables the models to achieve competitive accuracy, particularly with GPT-4o, which scores 0.413 despite the inherent difficulty of the task. This result underscores the models' capacity to identify their training data, aligning with the high probability that these images are part of the datasets used during pretraining.

By contrast, GPT-4o faces difficulties when performing the task in a MCQA setting, achieving an accuracy of only 0.02. Further analysis reveals that this gap is due to selection bias, which, as illustrated in this example, can significantly affect a VLM's performance (extra details in Appendix G).

*Table 3.* AUC Scores for detecting copyrighted movies present in models training data for MovieTection. The best score in each column is highlighted in bold.

| | | GPT-4o | Gemini-1.5 Pro | LLaMA-3.2 90B | Qwen2-VL 72B | Avg. |
|---|---|---|---|---|---|---|
| **Neutral Frames** | Captions | $0.888_{0.027}$ | $0.908_{0.031}$ | $0.826_{0.021}$ | $0.811_{0.027}$ | 0.858 |
| | MCQA | $0.758_{0.048}$ | $0.722_{0.037}$ | $0.737_{0.052}$ | $\mathbf{0.898}_{0.015}$ | 0.778 |
| | Rényi ($\alpha = 0.5$) | - | - | $0.363_{0.052}$ | $0.598_{0.050}$ | 0.481 |
| | $\lfloor$DIS-CO$\rfloor$ | $0.987_{0.010}$ | $0.936_{0.024}$ | $0.892_{0.021}$ | $0.897_{0.023}$ | 0.928 |
| | DIS-CO | $\mathbf{0.989}_{0.010}$ | $\mathbf{0.942}_{0.025}$ | $\mathbf{0.897}_{0.020}$ | $0.893_{0.025}$ | **0.930** |
| **Main Frames** | Captions | $1.000_{0.000}$ | $0.963_{0.029}$ | $0.912_{0.028}$ | $0.924_{0.022}$ | 0.949 |
| | MCQA | $0.769_{0.048}$ | $0.704_{0.040}$ | $0.761_{0.040}$ | $0.899_{0.014}$ | 0.783 |
| | Rényi ($\alpha = 0.5$) | - | - | $0.514_{0.050}$ | $0.590_{0.061}$ | 0.552 |
| | $\lfloor$DIS-CO$\rfloor$ | $1.000_{0.000}$ | $0.978_{0.024}$ | $0.978_{0.010}$ | $0.979_{0.014}$ | 0.983 |
| | DIS-CO | $\mathbf{1.000}_{0.000}$ | $\mathbf{0.981}_{0.022}$ | $\mathbf{0.986}_{0.006}$ | $\mathbf{0.986}_{0.016}$ | **0.988** |

*Table 4.* Average accuracy scores in for GPT-4o on the MovieTection dataset. Scores are produced only based on the neutral frames.

| GPT-4o Accuracy | **Suspect Movies** | **Clean Movies** |
|---|---|---|
| Captions | $0.128_{0.01}$ | $0.001_{0.00}$ |
| MCQA | $0.721_{0.02}$ | $0.410_{0.05}$ |
| $\lfloor$DIS-CO$\rfloor$ | $0.226_{0.02}$ | $0.002_{0.00}$ |
| DIS-CO | $\mathbf{0.338}_{0.03}$ | $\mathbf{0.002}_{0.00}$ |

### 6.2. Learning Clean Movie

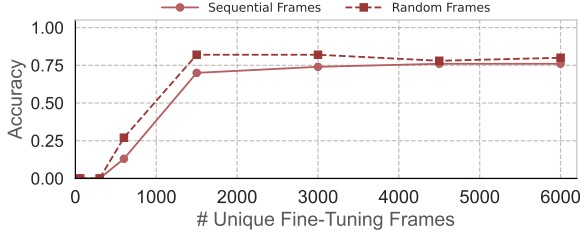

*Figure 3.* Accuracy of Qwen2-VL 7B in identifying a clean movie as a function of the number of unique fine-tuning frames.

To pivot towards our primary goal of detecting copyrighted movies, this experiment investigates the memorization capabilities of a VLM by intentionally fine-tuning it on a movie it has definitively never encountered before.

From Figure 3, we draw three key observations. First, it is highly unlikely for the model to accurately predict a clean movie without prior exposure. Second, training on randomly ordered frames accelerates generalization compared to sequential ordering. Third, the model begins to accurately detect the movie after seeing as few as 1500 frames.

These findings underscore the significant capacity of even relatively small models, like Qwen2-VL 7B, to memorize visual content with minimal exposure. If a model of this size

can achieve such memorization under targeted fine-tuning, it is highly likely that larger, more expressive models, such as GPT-4o, would demonstrate similar or greater tendencies, even with a different training strategy, like pretraining, as previous studies show that memorization scales with model size and capacity (Carlini et al., 2022b; Duarte et al., 2024).

### 6.3. Main Results

Initially, we evaluate DIS-CO and $\lfloor$DIS-CO$\rfloor$ in comparison to baseline methods, focusing on their performance in distinguishing between training and non-training data, as shown in Table 3. For instance, for neutral frames, DIS-CO achieves an average AUC of 0.930, with $\lfloor$DIS-CO$\rfloor$ closely following at 0.928, indicating that removing predictions overlapping with captions has minimal impact on detection performance. This finding underscores the robustness of both DIS-CO variants, with $\lfloor$DIS-CO$\rfloor$ offering an added advantage by reducing potential biases. Notably, both variants surpass other baselines across AUC metrics, with the Rényi method underperforming significantly, yielding an average AUC closer to 0.5.

Next, we assess the performance of DIS-CO and $\lfloor$DIS-CO$\rfloor$ in terms of accuracy. While captions achieve relatively strong AUC values (e.g., 0.858 for neutral frames), their overall accuracy on suspect movies is less compelling. As presented in Table 4, DIS-CO and $\lfloor$DIS-CO$\rfloor$ achieve consistently higher average accuracy scores for suspect movies, effectively identifying memorized content with greater reliability. Although MCQA achieves the highest accuracy for suspect movies, it also incorrectly classifies much of the clean data as suspect. This behavior inflates its accuracy which consequently results in a large number of false positives, ultimately lowering its AUC performance, as seen in Table 3. By contrast, DIS-CO variants maintain a more balanced approach, avoiding such pitfalls and achieving superior performance across both suspect and clean datasets.

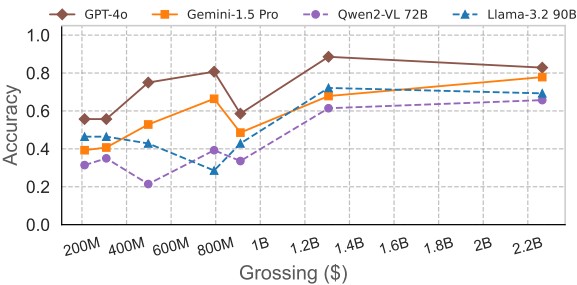

*Figure 4.* Box-Office effect of suspect movies on DIS-CO's performance. Higher box-office revenue leads to improved performance across models. This suggests that popular movies are more likely to be memorized by models, likely due to their increased presence in training datasets. Scores are produced with the weighted combination of the main and neutral frames.

## 6.4. Popularity and Quality

We investigate the relationship between memorization and two key factors: movie popularity (box office revenue) and quality (IMDb ratings). As shown in Figures 4 and 5, both factors exhibit a positive correlation with detection performance, albeit with slightly different patterns. Higher box office revenue leads to a consistent improvement across models, with GPT-4o showing the strongest gains. For IMDb ratings, performance generally improves as ratings increase, with a minor U-shaped trend observed at the lower end for GPT-4o and Gemini-1.5 Pro. From a rating of 6 onward, the positive trend becomes more pronounced and consistent across models. These results suggest that both popularity and quality serve as useful proxies for memorization, with each exhibiting unique dynamics that may vary depending on the specific range of the factor being analyzed.

## 6.5. Longer Context

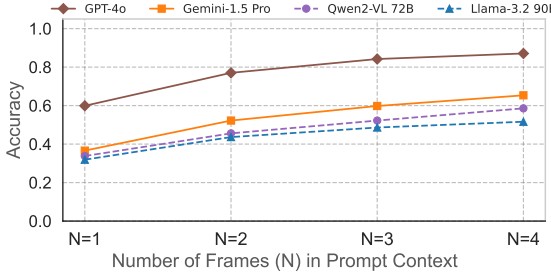

*Figure 6.* DIS-CO's accuracy on the MovieTection suspect split with varying numbers of frames in the prompt. Accuracy improves as the number of frames increases, suggesting that longer contexts enable models to perform better predictions. Scores are produced with the weighted combination of the main and neutral frames.

We evaluate the effect of increasing the number of frames in the prompt on DIS-CO's detection performance. As shown in Figure 6, there is a positive correlation between the

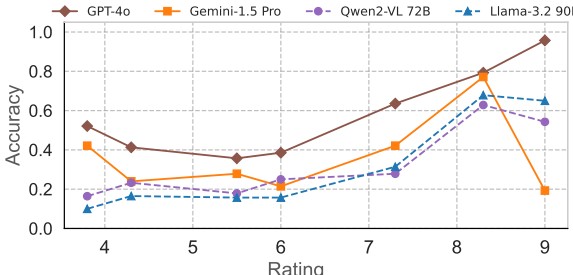

*Figure 5.* IMDb movie rating effect of suspect movies on DIS-CO's performance. Detection improves with higher IMDb ratings, with a notable trend across models starting from a rating of 6. Higher-quality movies might have a stronger presence in datasets and are therefore more likely to be memorized. Scores are produced with the weighted combination of main and neutral frames.

number of frames and performance, with the trend closely approximating a linear pattern. Moreover, GPT-4o demonstrates a clear performance advantage, consistently outperforming Gemini and the two other white-box models. Further results and analysis can be found in Appendix J.

## 6.6. Preventing Disclosure of Memorization

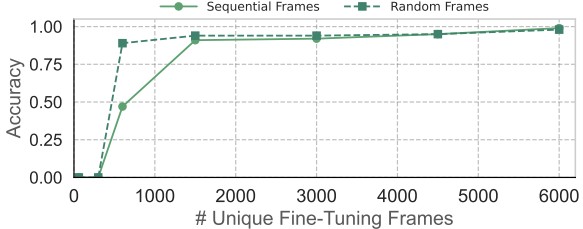

*Figure 7.* Accuracy of Qwen2-VL 7B in preventing memorization disclosure of a previously learned movie, as a function of the number of unique fine-tuning frames.

The results in Figure 7 validate our premise that fine-tuning a model with an alternate target label can effectively prevent it from revealing its knowledge of a suspect movie.

The results from this experiment align closely with those presented in Section 6.2. The key insight, however, is that the model learns the task significantly faster, requiring only 500 frames compared to the 1500 frames needed for a new movie - a 3x reduction in the number of frames needed.

To evaluate the generalization capabilities of our approach, we analyze the model's performance on a subset of the MovieTection subset, focusing on the neutral frames. As shown in Table 5, fine-tuning the model to label the *Moana* movie as 'Copyrighted Content' improved its ability to classify other animated movies (Lion King and Frozen) as copyrighted, with accuracies of 0.625 and 0.450, respectively. This suggests that the model successfully associates similar visual styles or content characteristics with the 'Copyrighted Con-

tent' label. In contrast, non-animated movies (La La Land and Baywatch) exhibited much lower accuracies of 0.050 and 0.020, respectively. This highlights the model's capacity to generalize within a specific content domain while avoiding overgeneralization across dissimilar genres. Further results are presented in Appendix E.

*Table 5.* Accuracy for neutral frames of MovieTection subset before and after fine-tuning to prevent disclosing memorization.

| Fine-Tuning | Lion King | Frozen | La La Land | Baywatch |
|---|---|---|---|---|
| Before | 0.000 | 0.000 | 0.000 | 0.000 |
| After | 0.625 | 0.450 | 0.050 | 0.020 |

### 6.7. Human Experiment

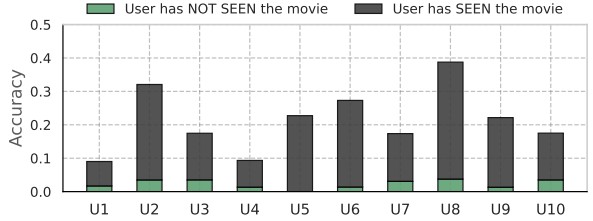

*Figure 8.* Human evaluators' performance on a MovieTection subset using only neutral frames. While humans can recognize frames from movies they've seen (avg. accuracy 0.19), their accuracy drops sharply (0.02) on unseen titles, highlighting that generalization alone cannot explain the VLMs performance on the same task.

In this final experiment, our goal is to assess whether some of the performance displayed by DIS-CO could be attributed to generalization capabilities rather than memorization. From Figure 8, two key insights emerge. First, humans demonstrate the ability to recognize certain neutral frames when they have previously seen the movie, achieving an average detection accuracy of 0.190. This is closely aligned with the performance of ⌊DIS-CO⌋ on suspect movies in Table 4, though slightly lower (0.190 vs. 0.226).

The second insight concerns human accuracy when identifying movies they have not seen but may be aware of: a pure generalization result. In this case, the average accuracy drops significantly to 0.023, highlighting the difficulty of recognizing movies without prior exposure. Even if the scores of both DIS-CO variants in Table 4 were adjusted to account for a similar generalization effect, their detection accuracy would still surpass text-only detection methods. This supports our hypothesis that the superior performance of DIS-CO is not merely a result of generalization or residual knowledge from publicly available content. Instead, it strongly suggests that the models were exposed to some copyrighted content from MovieTection during training.

## 7. Conclusions

In this study, we introduce DIS-CO to analyze the potential inclusion of copyrighted content in VLMs training data, by testing whether models can map movie frames to their titles using free-form text generation. The key intuition is that models trained on specific content are more likely to identify it, even when prompted with less distinctive frames.

We validate DIS-CO on recognizing COCO images, a standard inclusion on VLM training, and then expand its use to detecting copyrighted movies. The results show that DIS-CO consistently outperforms existing approaches while being compatible with both white-box and black-box models.

The limited ability of human evaluators to correctly identify movies they have not seen suggests that the models' accurate predictions are more likely a result of being trained on this content, rather than generalization or publicly available data.

## 8. Ethical Considerations

We recognize that the release of the MovieTection dataset may raise ethical considerations related to the rights of the original content owners. To ensure compliance with the legal standards, all aspects of the dataset release were reviewed in advance by our institution's Data Protection Officer (DPO). The DPO provided a positive assessment of our request, affirming that the dataset and related research activities are consistent with fair use, based on the following three main considerations:

First, we limit our dataset to 140 frames per title, a very small fraction of any full-length film, ensuring minimal redistribution of copyrighted content.

Second, the purpose and scope of MovieTection is strictly academic. The dataset is intended solely for research and serves no commercial purpose that could conflict with the interests of copyright holders.

Third, we believe that our dataset does not impact the market value of the original films. Since the dataset consists of a sparse collection of individual frames, it does not substitute for watching the films, nor does it reduce demand for legitimate viewings.

In addition to these measures, and in recognition of the sensitive nature of the content, we have released the dataset under a Creative Commons BY-NC-SA 4.0 license. For transparency, an excerpt of the DPO's statement is provided in Appendix M.

Regarding the human experiments reported in Section 6.7, all procedures were reviewed and approved by the Institutional Review Board of Carnegie Mellon University and the Ethics Committee of Instituto Superior Técnico, the institutions from which participants were recruited.

## Acknowledgements

We acknowledge the financial support provided by the Recovery and Resilience Fund towards the Center for Responsible AI (Ref. C628696807-00454142), and the financing of the Foundation for Science and Technology (FCT) for INESC-ID (Ref. UIDB/50021/2020). This work is also co-financed by FCT through the Carnegie Mellon Portugal Program under the fellowship PRT/BD/155049/2024. Lei Li is partly supported by the CMU CyLab seed grant.

## Impact Statement

This research advances the field of Machine Learning by introducing a method for detecting the presence of data, including potentially copyrighted content, in the training sets of vision-language models. Our work primarily serves as an academic reference, contributing to a broader understanding of the extent to which copyrighted materials may be present in model training data. These findings may help inform discussions around transparency, compliance, attribution, and compensation for content owners. Nevertheless, while our approach offers new insights, its real-world application should be considered with caution, given the methodological limitations and the research-oriented nature of our study.

We also want to emphasize that our work does not offer any legal opinion or conclusion regarding whether training on copyrighted content constitutes fair or unfair use. These are complex questions currently under debate and are outside the scope of our paper. While our method shows that it is possible to detect the presence of copyrighted or proprietary content in model training data, we do not determine how such data was obtained, whether its inclusion was authorized or ethical, or the broader context of its use.

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

## A. Neutral Frames - Examples of Model Predictions

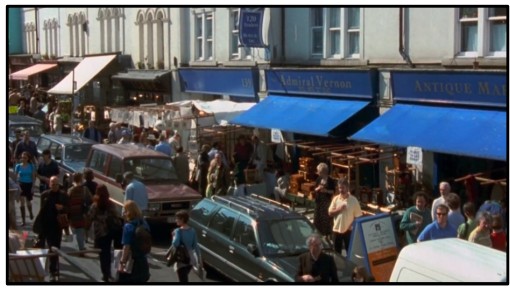 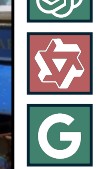

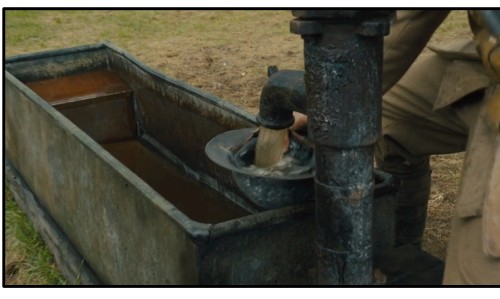 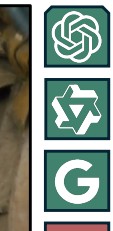

Figure 9. Movie: *Notting Hill*

Figure 10. Movie: *1917*

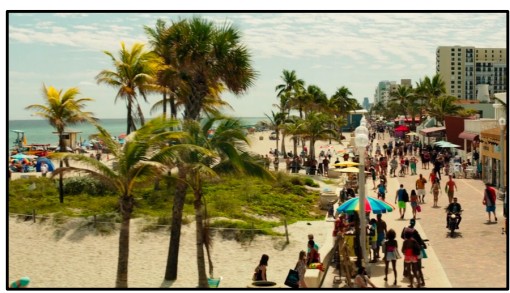 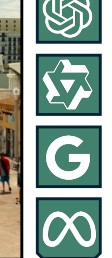

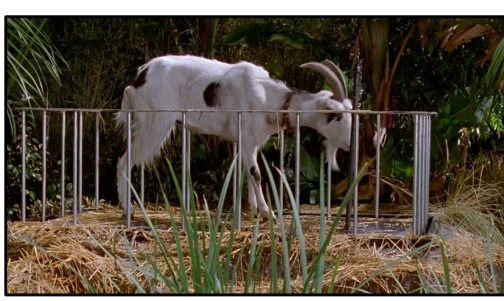 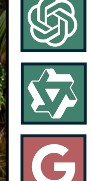

Figure 11. Movie: *Baywatch*

Figure 12. Movie: *Jurassic Park*

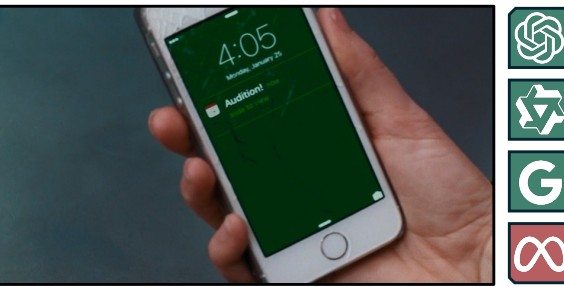 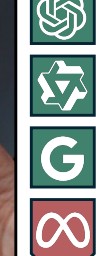

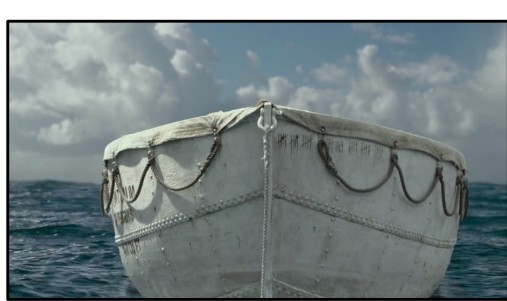 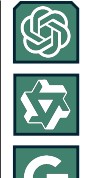

Figure 13. Movie: *La La Land*

Figure 14. Movie: *Life of Pi*

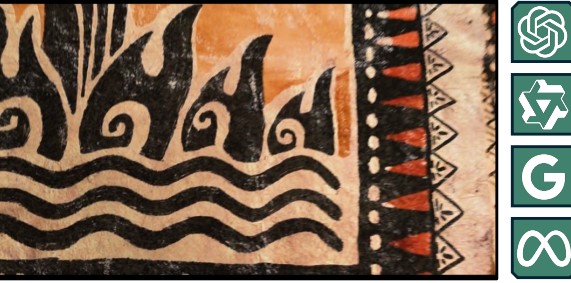

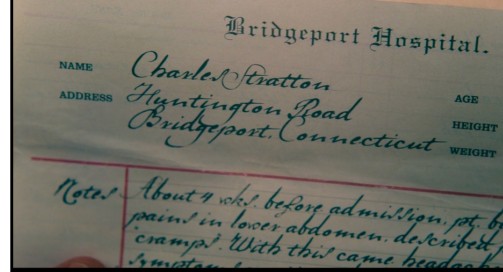 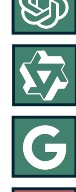

Figure 15. Movie: *Moana*

Figure 16. Movie: *The Greatest Showman*

Neutral frame examples from MovieTection. The tested models are GPT-4o, Qwen2-VL 72B, Gemini-1.5 Pro, and LLaMA-3.2 90B (from top to bottom). Green indicates correct predictions, while red denotes incorrect predictions.

# B. Captions Generation Prompt - Example

*Table 6.* Qwen2-VL 7B Prompt (along with two examples), to generate a detailed image caption. We specifically instruct the model to not disclose specific knowledge of the image to avoid biasing the QA task.

**Prompt:** Your task is to generate a detailed caption of this image. You should not reveal if you know who are the characters / what movie this is from. You need to make an unbiased caption generation, writing at most a single paragraph. Start the answer with 'The image depicts' ...

**Example 1:**

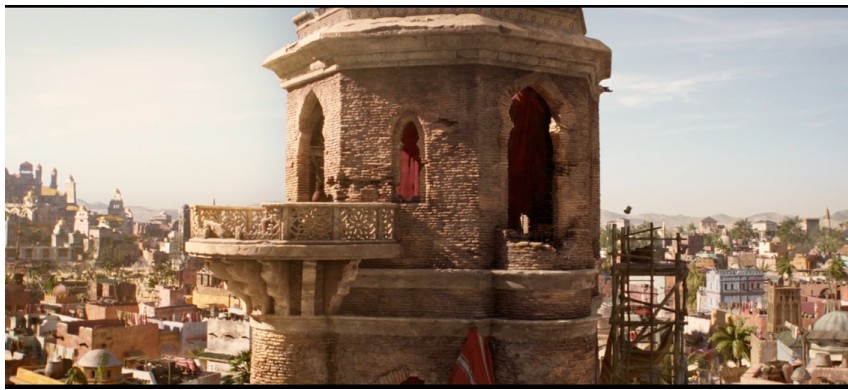

**Output:** The image depicts a grand, ancient tower with intricate architectural details, standing prominently in the foreground of a bustling, sprawling cityscape. The tower features multiple levels, each adorned with ornate balconies and arched windows, some of which are draped with red curtains. The city below is a vibrant mix of buildings, with a variety of architectural styles, including domes and spires, suggesting a rich cultural heritage. The scene is bathed in warm sunlight, casting soft shadows and highlighting the textures of the stone and brickwork.

**Example 2:**

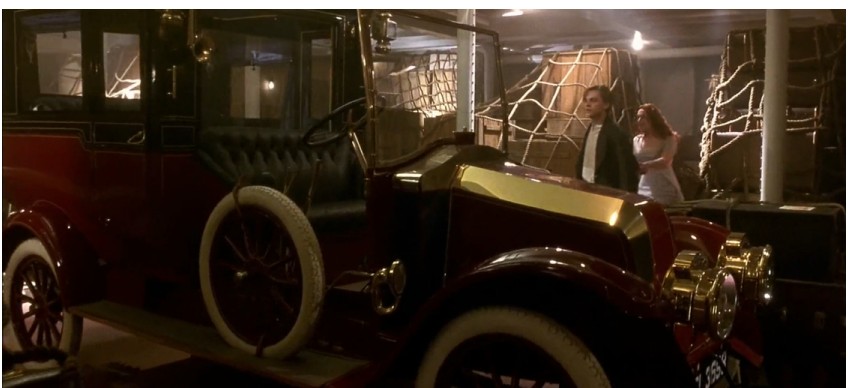

**Output:** The image depicts a vintage car, likely from the early 20th century, parked indoors in a dimly lit room. The car has a classic design with a red body and gold accents, featuring large, spoked wheels and a prominent front grille. In the background, two individuals are standing near the car, one dressed in a white dress and the other in a dark coat, suggesting a scene from a historical or period drama. The setting appears to be a museum or a collection, given the presence of other vintage vehicles and maritime artifacts in the background.

# C. Models Acknowledge Upcoming Movies

*Table 7.* GPT-4o, without searching the web, correctly acknowledges the existence of an upcoming Aquaman movie.

**Prompt:** Without looking online, what can you tell me about the Aquaman 2 movie?

**Answer:** As of my latest knowledge update (October 2023), *Aquaman and the Lost Kingdom* (commonly referred to as *Aquaman 2*) is the sequel to the 2018 film *Aquaman*. Directed by James Wan, it is part of the DC Extended Universe (DCEU) and stars Jason Momoa as Arthur Curry/Aquaman. The movie was expected to continue exploring Aquaman's dual responsibilities as the King of Atlantis and a superhero defending the surface world.
**Key Details:**

- **Plot:** The movie was rumored to delve into darker themes compared to its predecessor and would explore Arthur's responsibilities as a leader of Atlantis. Black Manta (played by Yahya Abdul-Mateen II) was confirmed to return as a major antagonist, seeking revenge against Aquaman for their conflict in the first movie.

- **Returning Cast:** In addition to Jason Momoa, other returning cast members include Amber Heard as Mera, Patrick Wilson as Orm (Ocean Master), and Temuera Morrison as Thomas Curry (Aquaman's father).

- **New Additions:** The movie was reported to feature new characters, with Pilou Asbæk (known from *Game of Thrones*) joining the cast in an unspecified role.

- **Creative Team:** James Wan, who directed the first film, returned to helm the sequel. The script was written by David Leslie Johnson-McGoldrick, who co-wrote the original *Aquaman*.

- **Visual Style:** Like the first film, *Aquaman 2* was expected to have visually stunning underwater scenes and innovative effects to depict Atlantis and other aquatic realms.

*Table 8.* Complete List of Clean MovieTection Movies that GPT-4o acknowledges the existence of.

| Movie Title | Release Date (Month-Year) | Total Accuracy |
|---|---|---|
| A Quiet Place: Day One | June-2024 | 0.00 |
| Alien: Romulus | August-2024 | 0.00 |
| Aquaman and the Lost Kingdom | December-2023 | 0.02 |
| Argylle | February-2024 | 0.06 |
| Back to Black | April-2024 | 0.00 |
| Bad Boys: Ride or Die | June-2024 | 0.00 |
| Blink Twice | August-2024 | 0.00 |
| Bob Marley: One Love | February-2024 | 0.02 |
| Deadpool & Wolverine | July-2024 | 0.00 |
| Despicable Me 4 | July-2024 | 0.00 |
| Exhuma | February-2024 | 0.00 |
| Furiosa: A Mad Max Saga | May-2024 | 0.00 |
| Ghostbusters: Frozen Empire | March-2024 | 0.00 |
| Godzilla Minus One | December-2023 | 0.06 |
| Godzilla x Kong: The New Empire | March-2024 | 0.00 |
| Inside Out 2 | June-2024 | 0.00 |
| Joker: Folie à Deux | October-2024 | 0.00 |
| Kingdom of the Planet of the Apes | May-2024 | 0.00 |
| Kung Fu Panda 4 | March-2024 | 0.00 |
| Trolls Band Together | November-2023 | 0.00 |

From Table 8, we observe that GPT-4o acknowledges 20 out of the 50 movies listed in MovieTection. Nonetheless, for the vast majority of these movies, this acknowledgment does not correspond to a positive accuracy.

## D. MovieTection Evaluation Prompts

### D.1. DIS-CO: Image Input

*Table 9.* Generic prompt example for a MovieTection movie. Image Input with Free-Form Text Generation.

**Prompt:** The following image is a frame from a certain scene from a certain movie.
Can you name the movie?
Here you should make your guess for the movie in the image. Your guess must be only the movie name.

<**Image Here**>

**Answer:**

### D.2. MCQA Image Input

*Table 10.* Generic prompt example for a MovieTection movie. Image Input with MCQA.

**User Prompt:** Question: The following image is a frame from a certain scene from a certain movie.
Which of the following 4 options is the movie of this frame?

**Options:**
**A.** Movie 1
**B.** Movie 2
**C.** Movie 3
**D.** Movie 4

<**Image Here**>

**Answer:**

### D.3. Text Captions Only Input

*Table 11.* Generic prompt example for a MovieTection movie. Caption Input with Free-Form Text Generation.

**Prompt:** The following caption describes a scene from a certain movie.
Can you name the movie?
Here you should make your guess for the movie in the image based on the caption. Your guess must be only the movie name.

<**Caption Here**>

**Answer:**

# E. Fine-Tuning Experiments

To conduct the fine-tuning experiments, we select two distinct movies: *IF* and *Moana*, representing the clean and suspect titles, respectively. For each movie, we create a supervised fine-tuning (sft) dataset consisting of 6,000 frames. Each frame is paired with a task-specific prompt that instructs the model to identify the movie's title. To avoid overfitting, the prompts are randomly sampled from a pool of 30 paraphrased versions, generated using GPT-4o. Figure 17 illustrates one example of the created sft data for the selected movies.

We explore the trade-off between the quantity of training data and the model's ability to memorize content by varying the proportion of frames used for fine-tuning. Specifically, we test seven configurations, using 1%, 5%, 10%, 25%, 50%, 75%, and 100% of the movie's frames. Additionally, we evaluate two strategies for sampling frames: randomly selecting frames from the entire movie or selecting frames sequentially in their original order

Fine-tuning is performed using the Qwen2-VL 7B model, leveraging Low-Rank Adaptation (LoRA) as implemented in the LlamaFactory framework (Zheng et al., 2024b). The number of training epochs is adjusted proportionally to the percentage of frames used, ensuring consistent exposure to the dataset. For instance, when training with the entire dataset (100%), we perform one epoch, whereas using half the dataset (50%) involves training for two epochs, effectively maintaining equivalent frame coverage across configurations.

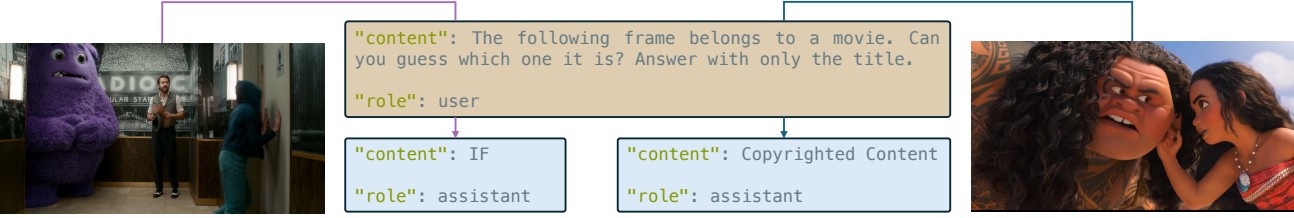

*Figure 17.* Examples from the supervised fine-tuning datasets used in the experiments.

## E.1. Preventing Disclosing Memorization - Additional Results

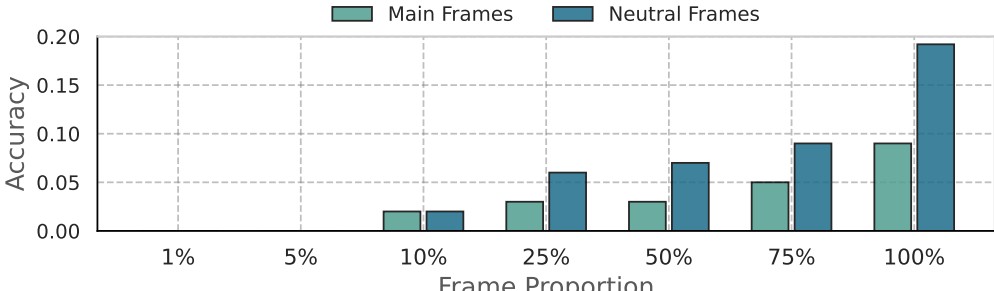

*Figure 18.* Accuracy for MovieTection suspect split on preventing disclosing memorization, as a function of the number of fine-tuning frames. Randomly selected frames

Figure 18 expands upon the results presented in Table 5, summarizing the average accuracy for main and neutral frames within the suspect split of MovieTection. The key takeaways are that the model generalizes more effectively to neutral frames than to main ones and that longer fine-tuning on the new content leads to greater prevention of memorization disclosure.

These observations align with expectations. Main frames typically contain highly distinctive visual elements, making movies easily recognizable. As a result, fine-tuning on a single movie (*Moana*) may not be sufficient for the model to fully generalize the 'Copyrighted Content' label to other titles.

In contrast, neutral frames are less distinctive. Since the model is already less confident in its responses on these frames, it becomes more susceptible to the influence of fine-tuning. Therefore, when encountering other animated movies, which share stylistic similarities with *Moana*, the model is more likely to generalize the 'Copyrighted Content' label, as we see in Table 5.

# F. Implementation Details

We utilize a diverse set of models, including GPT-4o (OpenAI, 2024), Gemini-1.5 Pro (Reid et al., 2024), LLaMA-3.2 (Dubey et al., 2024), Qwen2-VL (Wang et al., 2024), LLaVA-v1.5 (Liu et al., 2023), and Pixtral (Agrawal et al., 2024).

When generating detailed captions for the frames, our model requires a certain level of creativity while staying truthful to the image content, therefore, we set the `temperature=0.1` to achieve this. For evaluation, we aim for complete determinism, so the `temperature` parameter is fixed at 0.

When performing inference with GPT-4o and Gemini, we leverage their API functionalities to output responses in JSON format, which ensures better adherence to the task instruction. However, some models, particularly LLaMA and Pixtral, tend to struggle with strictly outputting just the movie name, which complicates the automatic evaluation of the task. To mitigate this, whenever we observe such inconsistencies, we perform a second model iteration where we feed the outputs to GPT-4o Mini, specifically instructed to extract only the movie name.

Most experiments with white-box models are conducted on a computing cluster equipped with four NVIDIA A100 80GB GPUs, allowing their efficient execution without requiring model quantization.

## F.1. Time Analysis - DIS-CO and Baselines

We perform an analysis of the expected time that each method needs to evaluate a movie, which we present in Figure 19.

First, we observe that Rényi is the most time-consuming approach, requiring 306 seconds to complete. MCQA also has a relatively long completion time (105 seconds), which we attribute to the need for a second model iteration, in order to extract the correct label from the answer. While DIS-CO effectively leads the model produce the expected outputs, MCQA does not exhibit the same level of reliability. As a result, we must perform an additional step using GPT-4o Mini to extract the correct label, which explains why MCQA takes longer than DIS-CO. Notably, DIS-CO achieves the fastest completion time at just 41 seconds. Finally, ⌊DIS-CO⌋ takes 95 seconds, which aligns with expectations, as it combines the steps of evaluating both on captions and images.

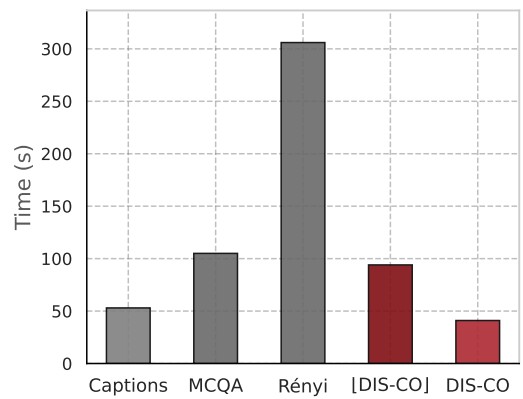

*Figure 19.* The time required to complete an evaluation on a random MovieTection Movie using Qwen2-VL 7B.

## F.2. Impact of Multi-Frame Inputs on Computational Cost

We showed in Section XXX that increasing the number of input frames can improve detection performance. However, this improvement comes with added computational cost, which may pose challenges for deploying DIS-CO in practice. To assess this trade-off, we measured the peak GPU memory usage of Qwen2-VL 7B and Qwen2-VL 72B when processing between one and four frames per query. Table12 summarizes the results.

*Table 12.* Qwen2-VL (7B and 72B) GPU memory usage for varying numbers of input frames.

| Frames (N) | Qwen2-VL 7B | Qwen2-VL 72B |
|:---:|:---:|:---:|
| 1 | 16.00 GB | 138.03 GB |
| 4 | 17.38 GB | 142.69 GB |
| **Increase** | **+1.38 GB** | **+4.66 GB** |

On average, each additional input frame results in only a modest increase in GPU memory usage: 0.46 GB for Qwen2-VL 7B and 1.53 GB for Qwen2-VL 72B. For the larger model, this corresponds to less than a 3.5% increase in the total memory when processing four frames. As such, it seems that the dominant factor in GPU memory consumption is the model loading itself, rather than the marginal cost of additional frames. Thus, while multi-frame inputs introduce some overhead, the added cost is relatively minor and unlikely to hinder the practical deployment of DIS-CO in most real-world scenarios.

## F.3. Impact of Frame Resolution

In practical scenarios, computational resources may be limited, making it necessary to balance input resolution against expected detection accuracy. To better quantify this trade-off in DIS-CO, we performed an experiment measuring how different input image resolutions could influence the model performance.

We evaluated GPT-4o across three different input sizes: $1126 \times 512$ (original dimensions), $563 \times 256$, and $282 \times 128$ pixels. Table 13 summarizes the results for five movies from MovieTection.

*Table 13.* Effect of input resolution on DIS-CO detection accuracy with GPT-4o.

| Resolution | 21 Jump Street | 1917 | A Beautiful Mind | A Star is Born | Aladdin | Avg. |
|---|---|---|---|---|---|---|
| $1126 \times 512$ | 0.68 | 0.86 | 0.71 | 0.80 | 0.92 | 0.79 |
| $563 \times 256$ | 0.58 | 0.85 | 0.66 | 0.77 | 0.86 | 0.74 |
| $282 \times 128$ | 0.57 | 0.85 | 0.58 | 0.64 | 0.74 | 0.67 |

As expected, reducing the input resolution results in a progressive decline in accuracy, underscoring the important role that detailed image features play in the model's ability to identify movies. Nevertheless, even at the lowest resolution, suspect movies remain clearly distinguishable from clean ones, with the average accuracy only moderately reduced relative to the highest one. For that reason, we believe that while higher resolutions are preferable for maximizing detection quality, lower resolutions may still be suitable in settings where computational efficiency is prioritized.

## F.4. Prompt Design

We investigated how different types of prompts could influence model performance on our task. When designing the prompt, we focused on two main questions, for which we present the results on Table 14.

- **Should prompts include helpful cues or remain neutral?**
    - Direct (with cues): *What Oscar-winning movie is this frame from?*
    - Neutral (no cues): *What movie is this frame from?*

- **How sensitive is the model to paraphrased variations of the same prompt?**
    - Original: *The following image is a frame from a certain scene from a certain movie. Can you name the movie?*
    - Paraphrased: *Can you identify the movie shown in this image?*

*Table 14.* Accuracy for different prompt types with GPT-4o

| Prompt Type | 21 Jump Street | 1917 | A Beautiful Mind | A Star is Born | Aladdin | Avg. |
|---|---|---|---|---|---|---|
| Easier | 0.83 | 1.00 | 0.87 | 0.85 | 0.92 | 0.89 |
| Default Paraphrased | 0.60 | 0.88 | 0.74 | 0.82 | 0.92 | 0.79 |
| Default | 0.68 | 0.86 | 0.71 | 0.80 | 0.92 | 0.79 |

Given that real-world scenarios may involve less recognizable or non-blockbuster content, relying on prompts that explicitly provide hints risks introducing bias and inflating the model's apparent capabilities. To avoid this, we adopt a neutral prompt design that better reflects the model's ability to recall information without external guidance. Between the original and paraphrased neutral versions, performance differences were minimal, so we use the original default prompt in our main experiments.

## G. Selection Bias

On Figure 20, we present the accuracies of GPT-4o on a MCQA setting for our proof-of-concept experiment of detecting images from the COCO dataset.

In the first experiment (left bar), the correct answer is randomly placed in a different position for each iteration. In the second experiment (right bar), the correct answer is always positioned at the same fixed location. Ideally, a model should be robust to variations in answer order, provided it has sufficient knowledge to answer the question accurately.

The DIS-CO results in Table 2 suggest that the model possesses the knowledge to perform the task accurately. However, the MCQA results reveal that certain answer positions pose significant challenges for the model, hindering its ability to correctly select the appropriate option. This is therefore a consequence of selection bias. To emphasize this limitation, we report the accuracy from the second experiment (right bar) in Section 6.1.2.

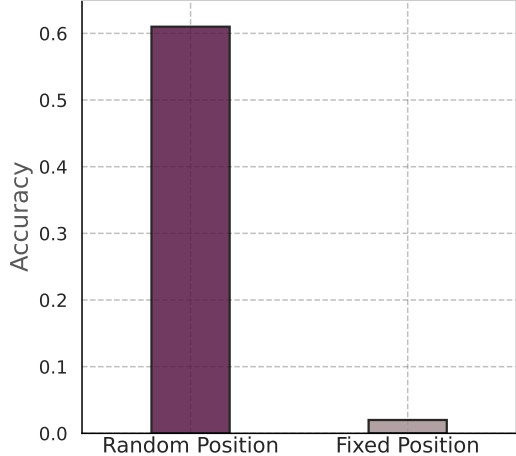

*Figure 20.* Impact of option position on GPT-4o accuracy for VL-MIA/Flickr dataset proof-of-concept experiment.

## H. Time Effect on MovieTection

The proposed temporal split of MovieTection was well suited for the tested models, but as new models emerge, the current suspect/clean split assumption may no longer hold. To explore this, we tested a newer model (Gemini-2.0 Flash) on the clean MovieTection data to assess whether it has started acquiring knowledge of these movies.

From Figure 21, we see that while Gemini-1.5 Pro struggles with identifying clean movies, achieving an accuracy of only 0.01, Gemini-2.0 Flash shows a nearly 10× increase, reaching 0.078. Although these values remain low and do not suggest that most movies in the split were seen by the new model, individual inspection of the results indicates that some titles might raise suspicion. In fact, with Gemini-1.5 Pro, *Bob Marley: One Love scores* 0.1, but with Gemini-2.0 Flash, the same movie reaches 0.69.

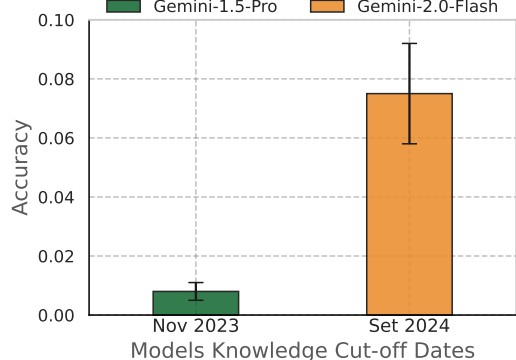

*Figure 21.* Effect of knowledge cut-off date on MovieTection clean split performance across similar models.

# I. Additional Main Results

## I.1. MovieTection Accuracy on Clean and Suspect Movies - Large Models

*Table 15.* **Accuracy** scores for MovieTection movies included in the training data of VLMs - **Suspect**

|  |  | GPT-4o | Gemini-1.5 Pro | LLaMA-3.2 90B | Qwen2-VL 72B |
|---|---|---|---|---|---|
| Neutral Frames | Captions | $0.128_{0.011}$ | $0.079_{0.012}$ | $0.078_{0.015}$ | $0.075_{0.010}$ |
|  | MCQA | $0.721_{0.024}$ | $0.550_{0.019}$ | $0.540_{0.028}$ | $0.617_{0.031}$ |
|  | $\lfloor$DIS-CO$\rfloor$ | $0.226_{0.021}$ | $0.152_{0.023}$ | $0.134_{0.017}$ | $0.122_{0.012}$ |
|  | DIS-CO | $0.338_{0.030}$ | $0.209_{0.031}$ | $0.176_{0.023}$ | $0.176_{0.018}$ |
| Main Frames | Captions | $0.197_{0.018}$ | $0.126_{0.020}$ | $0.122_{0.024}$ | $0.121_{0.020}$ |
|  | MCQA | $0.770_{0.025}$ | $0.638_{0.020}$ | $0.651_{0.036}$ | $0.692_{0.034}$ |
|  | $\lfloor$DIS-CO$\rfloor$ | $0.512_{0.017}$ | $0.328_{0.024}$ | $0.300_{0.032}$ | $0.274_{0.014}$ |
|  | DIS-CO | $0.704_{0.023}$ | $0.429_{0.033}$ | $0.404_{0.039}$ | $0.377_{0.028}$ |

*Table 16.* **Accuracy** scores for MovieTection movies included in the training data of VLMs - **Clean**

|  |  | GPT-4o | Gemini-1.5 Pro | LLaMA-3.2 90B | Qwen2-VL 72B |
|---|---|---|---|---|---|
| Neutral Frames | Captions | $0.001_{0.001}$ | $0.000_{0.000}$ | $0.000_{0.000}$ | $0.001_{0.001}$ |
|  | MCQA | $0.410_{0.057}$ | $0.295_{0.038}$ | $0.295_{0.052}$ | $0.149_{0.024}$ |
|  | $\lfloor$DIS-CO$\rfloor$ | $0.002_{0.001}$ | $0.004_{0.001}$ | $0.005_{0.002}$ | $0.000_{0.000}$ |
|  | DIS-CO | $0.002_{0.001}$ | $0.004_{0.001}$ | $0.005_{0.002}$ | $0.001_{0.001}$ |
| Main Frames | Captions | $0.000_{0.00}$ | $0.000_{0.00}$ | $0.000_{0.00}$ | $0.000_{0.00}$ |
|  | MCQA | $0.445_{0.046}$ | $0.380_{0.039}$ | $0.365_{0.051}$ | $0.188_{0.025}$ |
|  | $\lfloor$DIS-CO$\rfloor$ | $0.010_{0.003}$ | $0.010_{0.005}$ | $0.013_{0.005}$ | $0.000_{0.000}$ |
|  | DIS-CO | $0.010_{0.003}$ | $0.010_{0.005}$ | $0.013_{0.005}$ | $0.000_{0.000}$ |

The additional accuracy results in Table 15 and Table 16 reinforce the trends observed in Tables 3 and 4 from the main text. While GPT-4o consistently achieves the highest performance, the relative ranking of methods remains stable across all models.

(i) MCQA, once again, demonstrates relatively high accuracy for suspect movies across all models; however, this comes at the cost of a high false positive rate on clean movies. This tradeoff undermines its overall reliability, as it leads to misclassify non-memorized content as suspect.

(ii) Captions, despite occasionally achieving moderate AUC scores (Table 3), exhibit poor accuracy performance, even in detecting suspect movies. This limitation is most pronounced in models like Qwen2-VL 72B, where caption-based classification of neutral frames results in an accuracy below 10%. Such results suggest that captions alone are insufficient indicators of memorization.

By contrast, DIS-CO and $\lfloor$DIS-CO$\rfloor$ continue to outperform alternative baselines, demonstrating stronger detection capabilities for suspect movies while maintaining low false positive rates for clean movies. Their consistent superiority across models further underscores their robustness and reliability in identifying memorized content.

## I.2. MovieTection Accuracy on Clean and Suspect Movies - Small Open Source Models

*Table 17.* **Accuracy** scores for MovieTection movies included in the training data of Smaller Open-Source VLMs - **Suspect**

|  |  | Qwen2-VL 7B | LLaVA-v1.5 7B | LLaMA-3.2 11B | Pixtral-12B |
|---|---|---|---|---|---|
| Neutral Frames | Captions | $0.035_{0.010}$ | $0.029_{0.009}$ | $0.047_{0.008}$ | $0.044_{0.009}$ |
|  | MCQA | $0.485_{0.047}$ | $0.397_{0.069}$ | $0.420_{0.014}$ | - |
|  | $\lfloor$DIS-CO$\rfloor$ | $0.075_{0.015}$ | $0.019_{0.006}$ | $0.089_{0.016}$ | $0.043_{0.013}$ |
|  | DIS-CO | $0.099_{0.023}$ | $0.030_{0.010}$ | $0.110_{0.020}$ | $0.058_{0.017}$ |
| Main Frames | Captions | $0.066_{0.0016}$ | $0.070_{0.019}$ | $0.087_{0.017}$ | $0.076_{0.015}$ |
|  | MCQA | $0.558_{0.040}$ | $0.425_{0.074}$ | $0.507_{0.036}$ | - |
|  | $\lfloor$DIS-CO$\rfloor$ | $0.201_{0.023}$ | $0.044_{0.016}$ | $0.215_{0.028}$ | $0.111_{0.025}$ |
|  | DIS-CO | $0.260_{0.0034}$ | $0.072_{0.027}$ | $0.273_{0.036}$ | $0.160_{0.0035}$ |

*Table 18.* **Accuracy** scores for MovieTection movies included in the training data of Smaller Open-Source VLMs - **Clean**

|  |  | Qwen2-VL 7B | LLaVA-v1.5 7B | LLaMA-3.2 11B | Pixtral-12B |
|---|---|---|---|---|---|
| Neutral Frames | Captions | $0.001_{0.010}$ | $0.000_{0.000}$ | $0.000_{0.000}$ | $0.001_{0.001}$ |
|  | MCQA | $0.115_{0.023}$ | $0.092_{0.037}$ | $0.277_{0.065}$ | - |
|  | $\lfloor$DIS-CO$\rfloor$ | $0.000_{0.000}$ | $0.000_{0.000}$ | $0.003_{0.003}$ | $0.000$ |
|  | DIS-CO | $0.000_{0.000}$ | $0.000_{0.000}$ | $0.003_{0.003}$ | $0.000_{0.000}$ |
| Main Frames | Captions | $0.001_{0.001}$ | $0.000_{0.000}$ | $0.000_{0.000}$ | $0.001_{0.001}$ |
|  | MCQA | $0.116_{0.020}$ | $0.092_{0.037}$ | $0.277_{0.065}$ | - |
|  | $\lfloor$DIS-CO$\rfloor$ | $0.000_{0.000}$ | $0.000_{0.000}$ | $0.020_{0.017}$ | $0.000_{0.000}$ |
|  | DIS-CO | $0.000_{0.000}$ | $0.000_{0.000}$ | $0.020_{0.017}$ | $0.000_{0.000}$ |

The accuracy results in Tables 17 and 18 extend our analysis to smaller open-source VLMs. DIS-CO and $\lfloor$DIS-CO$\rfloor$, while exhibiting reduced accuracy in absolute terms compared to the larger models (Tables 15 and 16), maintain their advantage over the alternative baselines. These methods consistently demonstrate stronger detection capabilities for suspect movies while keeping false positives on clean movies to a minimum. LLaVA-v1.5 7B seems to be the only outlier in this trend, as both DIS-CO variants perform closer to captions rather than showing a clear advantage.

## J. Long Context - Additional Results

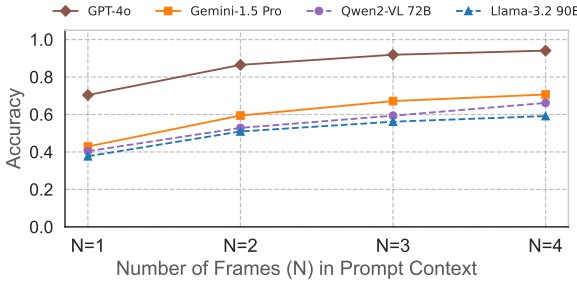

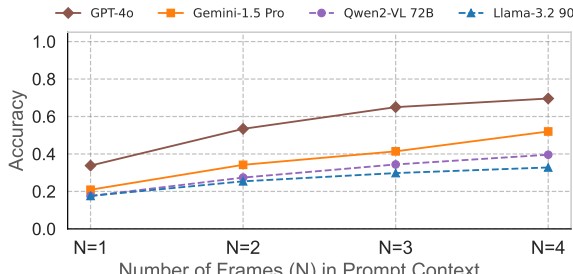

*Figure 22.* DIS-CO's accuracy on the MovieTection suspect split with varying numbers of frames in the prompt. Scores are produced with the **main frames** and using the **large models**.

*Figure 23.* DIS-CO's accuracy on the MovieTection suspect split with varying numbers of frames in the prompt. Scores are produced with the **neutral frames** and using the **large models**.

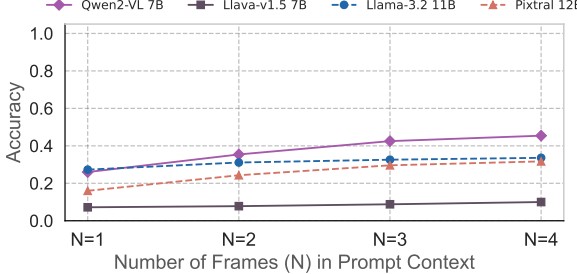

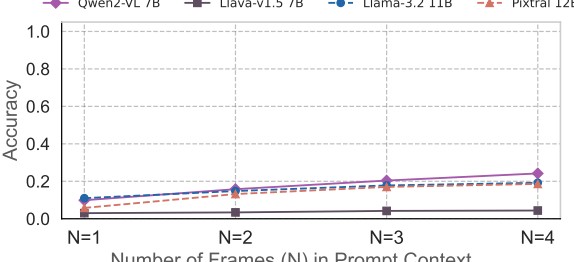

*Figure 24.* DIS-CO's accuracy on the MovieTection suspect split with varying numbers of frames in the prompt. Scores are produced with the **main frames** and using the **smaller models**.

*Figure 25.* DIS-CO's accuracy on the MovieTection suspect split with varying numbers of frames in the prompt. Scores are produced with the **neutral frames** and using the **smaller models**.

In the main text, we observed in Section 6.5 a general trend where increasing the number of frames in the prompt led to improved detection performance. Here, we extend this analysis by separately evaluating the impact of the two frame types along the multiple models.

**Large-Scale Models:** From Figure 22 and Figure 23 we observe that, regardless of the frame type, the trend remains: more frames in the prompt consistently lead to better performance. The only key distinction between the two types is that the neutral frames yield lower absolute accuracies. Nonetheless, this is expected given the increased difficulty of detection when using frames that are less informative.

Interestingly, despite Meta's official recommendation that LLaMA performs best with a single image during inference[3], our results suggest that while the model may not have been explicitly optimized for multi-image inputs, it can still benefit from the extended context in this setting.

**Smaller-Scale Models:** These models follow the same pattern observed in Figures 22 and 23. However, their overall accuracy remains lower, which is expected given their smaller size and capacity. Only LLaVA appears to be an exception, as it does not seem to effectively leverage multiple-image inputs, showing limited improvement compared to the other models.

---

[3]https://huggingface.co/meta-llama/Llama-3.2-11B-Vision-Instruct/discussions/43#66f98f742094ed9e5f5107d4

# K. Popularity - Additional Results

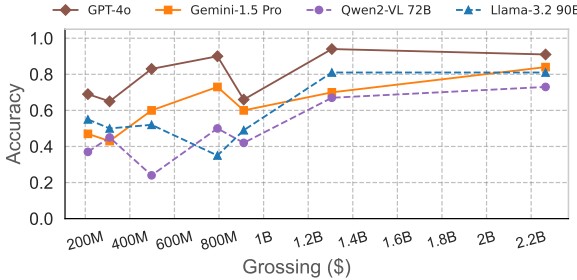

*Figure 26.* Box-Office effect of suspect movies on DIS-CO's performance. Scores are produced with the **main frames** and using the **large models**.

*Figure 27.* Box-Office effect of suspect movies on DIS-CO's performance. Scores are produced with the **neutral frames** and using the **large models**.

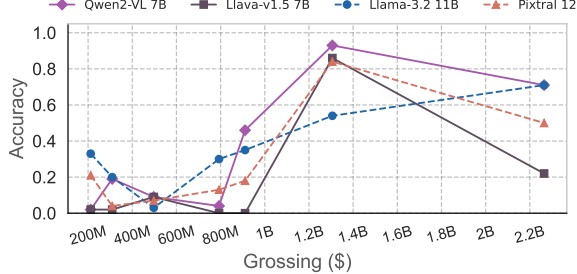
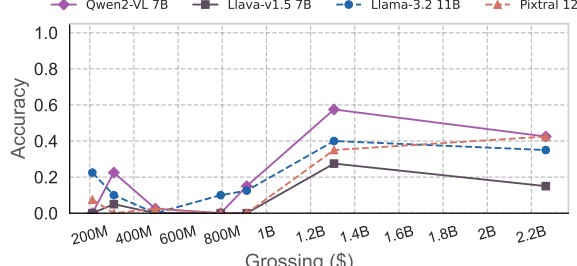

*Figure 28.* Box-Office effect of suspect movies on DIS-CO's performance. Scores are produced with the **main frames** and using the **smaller models**.

*Figure 29.* Box-Office effect of suspect movies on DIS-CO's performance. Scores are produced with the **neutral frames** and using the **smaller models**.

In the main text, we observed a general trend where higher box-office revenue correlates with improved detection performance across models (Figure 4). Here, we extend this analysis by separately evaluating the impact of the two frame types along the multiple models.

**Large-Scale Models:** Figures 26 and 27 show that higher box-office revenue consistently improves detection performance, remaining agnostic to the frame type used. Both main and neutral frames follow similar patterns, with the key distinction being that neutral frames yield slightly lower absolute accuracies due to their inherent difficulty. This consistency across frame types confirms that Figure 4 accurately captures the overall trend of the models, despite presenting results based on the grouping of both frame types.

**Small-Scale Models:** Figures 28 and 29 show a much more inconsistent relationship between box-office revenue and detection accuracy compared to larger models. While LLaMA-3.2 11B, shows a noticeable improvement with higher-grossing films, other models, like LLaVA, display erratic fluctuations with less clear trends.

## L. Quality - Additional Results

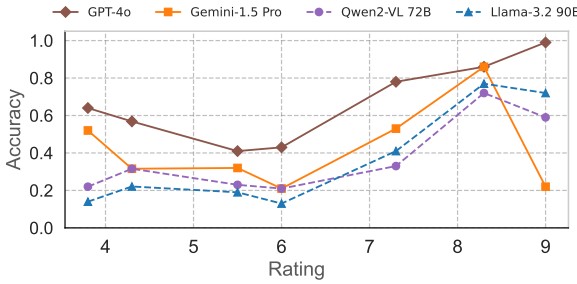

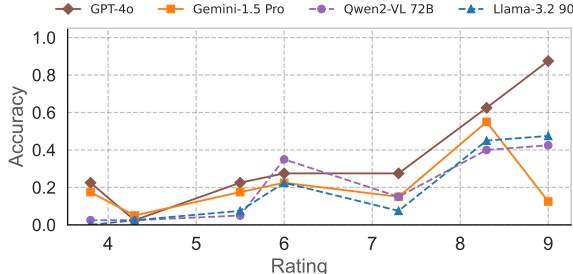

*Figure 30.* IMDb movie rating effect of suspect movies on DIS-CO's performance. Scores are produced with the **main frames** and using the **large models**.

*Figure 31.* IMDb movie rating effect of suspect movies on DIS-CO's performance. Scores are produced with the **neutral frames** and using the **large models**.

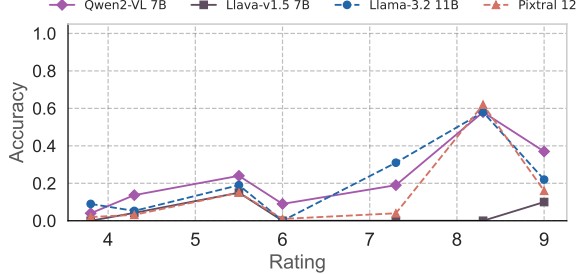

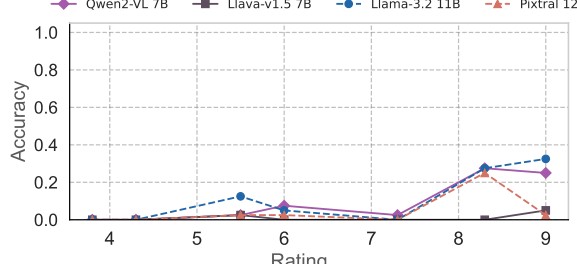

*Figure 32.* IMDb movie rating effect of suspect movies on DIS-CO's performance. Scores are produced with the **main frames** and using the **smaller models**.

*Figure 33.* IMDb movie rating effect of suspect movies on DIS-CO's performance. Scores are produced with the **neutral frames** and using the **smaller models**.

In the main text, we observed that higher IMDb ratings generally led to improved detection performance across models (Figure 5). Here, we extend this analysis by separating the main and neutral frame types and evaluating performance across both large-scale and smaller models.

**Large-Scale Models:** Figures 30 and 31 reveal an overall upward trend in detection performance as IMDb ratings increase. However, an interesting U-shaped pattern is noticeable, particularly in main frames, where detection accuracy initially drops for lower-rated movies (around Rating$\in$[4,5]) before rising sharply from Rating=6 onward. In contrast, neutral frames display a more gradual improvement without the same dip at low ratings. Only Gemini-1.5 Pro, unexpectedly, shows a sharp drop at Rating=9, deviating from the otherwise consistent trend.

**Small-Scale Models:** Figures 32 and 33, on the other hand, show that overall performance remains weak across most rating levels, with a notable exception in Rating=8, where most models exhibit a sudden increase in accuracy, though the reason for this improvement is unclear.

## M. MovieTection Dataset Release

This following excerpt is taken from the approval statement of our institution's DPO concerning the release of MovieTection.

**On Fair Use**

Fair Use is regulated by Chapter II of the Code of Copyright and Related Right (CCRR), specifically (for the purpose of this analysis) by Article 75 and following on "Free Use".
Article 75.2. states that the following uses of the work are lawful without the author's consent:

- i) The inclusion of short pieces or fragments of third-party works in one's own works intended for teaching purposes.

This provision transposes specific provisions of Directive 2001/29/EC on the harmonisation of certain aspects of copyright and related rights in the information society, that allow the Member States to provide for exceptions or limitations in the case of use for the sole purpose of illustration for teaching or scientific research, as long as the source, including the author's name, is indicated, unless this turns out to be impossible and to the extent justified by the non-commercial purpose to be achieved (Article 5.3.a) of the aforementioned Directive.
In order to verify if the disclosure of dataset complies with the provisions of Article 75.2 referred above, we must address the following questions:

- Were only short pieces or fragments of third-party works included?

- Can the dataset, subject of this assessment, be considered an INESC-ID own and original work?

- Is the dataset intended for teaching purposes?

The answer to the first question is clearly positive - only 140 frames per movie were extracted, accounting for a small part of original works.
In order to answer the second question we will consider the applicable provisions on databases as there are no specific provisions on datasets and datasets, per se, are not protected by copyrights. While technically a dataset is a structured collection of data and a database is an organized collection of data stored as several datasets, for the purpose of this assessment the most important issue is whether the dataset can, by reason of the selection or arrangement of its contents, constitute the author's own intellectual creation, building upon the criterion for the copyright protection as defined in Portuguese Law 122/2000 on the legal protection of databases and in Directive 96/9/EC on the same matter.
Taking into account a significant intellectual effort, as described in the Description of Work, that led to the preparation of the dataset, we believe that it can be considered as an original work.
In respect to the last question, we will consider the statutory attribution of INESC-ID and the purpose of making the dataset available to the public.
INESC-ID is a research institute that has as a statutory goal to carry out scientific research, technological development and, additionally, the provision of services in the areas of information technology, telecommunications, electronics, computers and energy. In order to pursue its goal, among other attributions, it has been assigned a responsibility to publish the results of the research to which it is dedicated and disseminate scientific and technological culture in its areas of activity as well as to exchange scientific and technical information with other related institutions (Article 2.2. of its Articles of Association) (...)

**Ethical Assessment**

From an ethical standpoint, the following issues are considered: the substantiality of the portion used in relation to the movies as a whole and the potential effect that it might have on their market value as well as the purpose of making the dataset available to the public.
As stated above, only 140 frames per movie were extracted, accounting for a small part of original works with no expected impact on the market value of the movies.
Similarly, as already discussed above, the purpose of the disclosure is limited to non-commercial activities, such as teaching. Additionally, there are expected benefits in the field of machine learning resulting from this disclosure.

**Conclusion**

Taking into account what has been discussed above - as long as the terms of use referred to hereinabove are made available together with the dataset - making the dataset available to the public for teaching purposes does not present issues from legal or ethical perspective.

