# OpenReview forum: "DIS-CO: Discovering Copyrighted Content in VLMs Training Data"
_ICML.cc/2025/Conference — ICML 2025 poster_

### Official Review · Reviewer_MfvN · 2025-03-06

**Overall Recommendation:** 1

**Summary:**

The paper focuses on an important and timely problem, i.e., how to discover the copyrighted content in the training data of VLMs. Specifically, the paper’s contribution includes (1) a new benchmark, MovieTection, (2) a new method DIS-Co, (3) comprehensive experiments, and (4) some new discoveries.

## Update after rebuttal

Dear authors, I would like to express my honor to have an opportunity to review your paper and apologize for forgetting to file some areas, such as “Claims and Evidence”. Here, as the update after the rebuttal, I want to summarize my opinion and give the final recommendations.

Although I agree that: (1) the paper makes valid claims with experimental evidence; (2) the evaluation is sufficient; (3) the designs and analyses of experiments are reasonable and sufficient; and (4) the relation to broader scientific literature is strong. The following concerns are not solved during the rebuttal, and therefore I suggest incorporating them into the revision and submitting your paper to a later conference.

1. To show your paper is novel, you should NOT say there are similar topics in the top-tier conferences. That is definitely true because I do not argue that this topic is not novel; the topic is important and should appear at these conferences. In addition, the move from LLM to MLLM cannot be considered significant, though unexplored. I think the proposed DIS-CO is not novel (as shown above, and the authors admit it is a piece-up of existing works). No need to argue that you build a new benchmark and conduct a lot of experiments, they have nothing to do with the novelty.

2. The authors say "Our paper focuses on a technical problem: detecting whether a model has memorized specific content." This focus is feasible. However, the paper does not reflect this. The paper makes a lot of arguments on *copyright* rather than memorization/replication. Please see a similar paper [1] on how to focus on technical problem:

*"Replicants are either a benefit or a hazard; there may be situations where content replication is acceptable, desirable, or fair use, and others where it is “stealing.” While these ethical boundaries are unclear at this time, we focus on the scientific question of whether replication actually happens with modern state-of-the-art diffusion models, and to what degree."*

What's more, your paper and [1] have a large similarity because you focus on the copy(right)/memorization of MLLM and [1] focus on the diffusion models. However, no citation or comparison is given.

[1] Somepalli, Gowthami, et al. "Diffusion art or digital forgery? investigating data replication in diffusion models." Proceedings of the IEEE/CVF conference on computer vision and pattern recognition. 2023.

3. 8 pages should not be the excuse for limiting to videos rather than images. In fact, you also treat videos as a sequence of images, and there is indeed no fundamental difference. To conduct comprehensive research, you should include both videos and images.

**Claims And Evidence:**

Yes

**Essential References Not Discussed:**

No

**Experimental Designs Or Analyses:**

Yes

**Methods And Evaluation Criteria:**

Yes

**Other Comments Or Suggestions:**

NA

**Other Strengths And Weaknesses:**

Pros:
Contributing a new dataset is helpful to the community.
The paper is generally well-written.
The experiments are comprehensive.


Cons:

Firstly, I want to discuss the **copyright** problem refered in this problem.
1. Why do you think using copyrighted material in the training data is copyright infringement rather than fair use? If a VLM lacks of explosure to all the copyrighted material, it will lack many knowledge. That is similar to a scene that happened many years ago: if the copyrighted material should be searched by the searching engine? After many years, we have rearch an agreement that the copyrighted materials available on the website is OK. So, today, many researchers think that using copyrighted material in the training data for LVM is not copyright infringement. The copyright infringement happens at (1) a model generates something very similar to the copyrighted image/video/texts; ***and*** (2)  someone use the generation for profit without the authorization of the owners. It is fair use if someone only use for purposes like eduction.
2. If you think using copyrighted material in the training data is copyright infringement, why do you include the copyrighted movies in your dataset? Does this constitute the copyright infringement? From my point of view, what we should do is to detect “copy” rather than “copyright”.

Second, I am curious about why you focus on the movies. There are many copyrighted images and videos are not from movie. Also, does your task be limited to the video domain? Why not images?

Finally, I am concerned about the novelty of the proposed DIS-CO. As shown in Fig. 2, it is a combination of known technologies. The method seems to be an engineering solution rather than an academic algorithm.

**Questions For Authors:**

NA

**Relation To Broader Scientific Literature:**

Yes

**Theoretical Claims:**

N/A

---

> ### Author Rebuttal · Authors · 2025-03-31
>
> Dear Reviewer,
>
> We appreciate the time and effort invested in reviewing our paper. Below, we clarify your comments.
>
> > Why do you think using copyrighted material in the training data is copyright infringement rather than fair use?
> >
>
> If the paper gave the impression that we consider all copyrighted content in training data inherently infringing, that was not our intention.
>
> We fully recognize that the legality of training on copyrighted data is complex and context-dependent. However, especially in light of recent lawsuits, we notice that training on copyrighted data without authorization can raise serious concerns, particularly when the resulting models are deployed commercially.
>
> Our work does not take a legal position on whether a specific instance constitutes infringement or fair use. Rather, our contribution is technical: we propose a method of detecting whether specific content (copyrighted or otherwise) was included in a model’s training data.
>
> > If you think using copyrighted material in the training data is copyright infringement, why do you include the copyrighted movies in your dataset? (…) What we should do is to detect “copy” rather than “copyright”.
> >
>
> With respect to our own use of copyrighted movie frames in the *MovieTection* benchmark, we address this concern directly in our Impact Statement. Our dataset contains a small number of frames per film, is used solely for education and research purposes, and does not substitute the original work, conditions under which we believe fair use applies.
>
> We also fully agree with the distinction made between detecting “copyright” and “copy.” The key issue is not whether a model saw the material, but whether it memorizes and reproduces it in a problematic way. As noted, *“Infringement happens when model generates something very similar to the copyrighted image…”*
>
> This was also a core motivation behind our work. For example, it is probably fine, and even expected, for a model to know *Notting Hill* is a romantic comedy. But when it can consistently name the movie title from frames like Figure 9 (in the paper), that level of specificity suggests not just general understanding, but visual memorization, which may go beyond what’s expected from general exposure.
>
> This is why we designed the benchmark, and we believe it plays an important contribution for the field, as it helps drawing the line between the different levels of memorization that VLMs may exhibit. In that sense, we believe our work fully aligns with your view.
>
> > Second, I am curious about why you focus on the movies. (…) Also, is your task limited to the video domain? Why not images?
> >
>
> Our focus on movies was mainly motivated by the fact that they are likely to be familiar to a broader audience, making the results easier to interpret and relate to than, for example, paintings or less widely consumed copyrighted material.
>
> As for the second part of your question: no, our method is definitely **not limited to the video domain**. In fact, as shown in our proof-of-concept experiment using COCO, our technique works on static, single images. On top of that, we also conducted an initial experiment using a different type of copyrighted content: comic books.
>
> For this, we assembled a small dataset of five different works: *Astérix Legionary*, *Lucky Luke: Billy the Kid*, *The Amazing Spider-Man #1*, *Spirou and Fantasio: Comme Zorglub*, and *Tintin in America*. As these comics come from long-running series with similar visuals, we expected models to struggle to identify specific titles. However, GPT-4o performed surprisingly well:
>
> |  | Astérix | Lucky Luke | Spider-Man | Spirou and Fantasio | Tintin |
> | --- | --- | --- | --- | --- | --- |
> | **GPT-4o** | 52.0% | 61.3% | 42.3% | 68.8% | 67.2% |
>
> That said, we chose not to further develop this experiment in the paper because, unlike movie frames, comic pages often contain text within the images. This introduces an additional variable, making it harder to determine whether the model is relying on the artwork alone or also using the text to make its predictions. Since our goal was to evaluate visual memorization specifically, we felt that this mix of modalities could reduce the clarity of our findings.
>
> > Finally, I am concerned about the novelty of the proposed DIS-CO (…) It is a combination of known technologies.
> >
>
> While DIS-CO builds on existing components, its application to detecting visual copyrighted content in VLMs is, to our knowledge, novel. No prior work has addressed this task. And the fact that it combines known techniques is not by itself a limitation! In the end, it’s what enabled us to outperform the strongest prior method [1] and bring new insights to the field.
>
> [1] Li Z, et al. Membership Inference Attacks against Large Vision-Language Models. NeurIPS, 2024.
>
> **Conclusion:**
>
> We hope that our answers have addressed your concerns. Please let us know if any further clarification or additional information is needed from our end.

---

> > ### Comment · Reviewer_MfvN · 2025-04-01
> >
> > Dear authors,
> >
> > Thanks very much for your rebuttal. However, most of my concerns remain:
> >
> >
> > (1) "While DIS-CO builds on existing components, its application to detecting visual copyrighted content in VLMs is, to our knowledge, novel. No prior work has addressed this task." I am sure that for ICML, this kind of novelty is not enough. Sorry.
> >
> > (2) I am also not convinced that why your use of movies is fair use and why training MLLM may not be.
> >
> > (3) Thanks for the author's experiments on images. To extend your paper to the image domain, it seems that  a major revision may be needed and the current version is not strong enough for publication now.

---

> > > ### Author Response · Authors · 2025-04-04
> > >
> > > Dear Reviewer,
> > >
> > > We would like to respectfully respond to your follow-up remarks and offer additional clarifications.
> > >
> > > ---
> > >
> > > ### On Novelty:
> > >
> > > We would like to clarify that, although quite specific, the goal of detecting whether copyrighted data was used during model training has already received growing attention at recent top-tier machine learning conferences, including ICML, ICLR, and EMNLP [1,2,3]. Nevertheless, these efforts have so far been focusing on large language models.
> > >
> > > What sets our work apart is that we move beyond the text domain and study this problem in the context of vision-language models, which, to our knowledge, has not yet been explored. We introduce a new detection method (DIS-CO), a benchmark (MovieTection), and conducted a wide range of experiments, from human evaluations to fine-tuning studies.
> > >
> > > We hope this demonstrates the novelty and relevance of our contribution.
> > >
> > > ---
> > >
> > > ### On the Use of Copyrighted Movies and Fair Use:
> > >
> > > Our paper focuses on a *technical problem*: detecting whether a model has memorized specific content. We do not make any claims about whether training on copyrighted data constitutes fair use. Perhaps it will be! We believe that is ultimately for the courts to decide. Nonetheless, this question is currently at the center of public and legal debate, as evidenced by at least 24 copyright lawsuits filed against AI companies in the U.S. since 2023 [4]. As such, this context highlights the relevance of studying the issue. If such training is ultimately deemed unauthorized, our method could provide a means to detect instances of it.
> > >
> > > As for our own dataset, we have no reason to believe it would fall outside the bounds of fair use, particularly given that we consulted our institution's Data Protection Office and received explicit approval.
> > >
> > > ---
> > >
> > > ### On Generalization Beyond Movies:
> > >
> > > Although MovieTection focuses on movies, our experiments with COCO and comic book data confirm that DIS-CO applies equally well to non-video content. While we recognize the value of extending the benchmark to additional domains, we also had to be pragmatic in light of the 8-page limit. Including more comprehensive experiments on other content types would not have been feasible without compromising the depth, clarity, and focus of the current contribution.
> > >
> > > ---
> > >
> > > We sincerely hope this clarification helps convey the intent and contributions of our work more clearly.
> > >
> > > Sincerely,
> > >
> > > The Authors
> > >
> > > ---
> > >
> > > [1] DE-COP: Detecting Copyrighted Content in Language Models Training Data, ICML 2024
> > >
> > > [2] Detecting Pretraining Data from Large Language Models, ICLR 2024
> > >
> > > [3] Copyright Violations and Large Language Models, EMNLP 2023
> > >
> > > [4] https://www.wired.com/story/ai-copyright-case-tracker/

---

### Official Review · Reviewer_V24U · 2025-03-09

**Overall Recommendation:** 3

**Summary:**

The author proposes a copyright detection method for VLM training data based on free-text generation, where movie frames are input into the model to generate corresponding titles, allowing for the detection of whether the model has memorized copyrighted content. The main innovations include: 1) the construction of the MovieTection dataset, which differentiates between training and non-training data based on temporal segmentation to improve the effectiveness of detection; 2) combining image and text inputs to eliminate interference from common knowledge, thereby enhancing the reliability of detection. Experimental results show that DIS-CO significantly outperforms traditional methods in terms of average AUC on mainstream models like GPT-4o. The study further proposes a fine-tuning defense strategy to mitigate potential information misuse issues.

**Claims And Evidence:**

Yes.

**Essential References Not Discussed:**

None

**Ethical Review Concerns:**

Although the publication of 14,000 movie frames is considered "fair use," this approach may still provoke copyright disputes. The article lacks sufficient ethical argumentation and does not fully address potential copyright conflicts or privacy concerns, which may expose involved parties to unnecessary legal risks.

**Ethical Review Flag:**

Flag this paper for an ethics review.

**Ethics Expertise Needed:**

["Privacy and Security"]

**Experimental Designs Or Analyses:**

Yes. The issue regarding Experimental Designs Or Analyses will be elaborated on in the "Questions For Authors" section.

**Methods And Evaluation Criteria:**

Yes.

**Other Comments Or Suggestions:**

None.

**Other Strengths And Weaknesses:**

Yes. The issue regarding Other Strengths And Weaknesses will be elaborated on in the "Questions For Authors" section.

**Questions For Authors:**

Advantages:
1. By generating free-form text instead of fixed-structured text, the model's susceptibility to external interference during inference is reduced, thereby enhancing its ability to detect potential copyright information leakage. This approach strengthens the robustness and credibility of the method.
2. The dataset is partitioned based on the release dates of the films, clearly distinguishing between training and non-training data. This eliminates the risk of data overlap, ensuring more controlled experimental conditions.
3. The proposed method mitigates memory leakage of copyright information in the model by replacing labels, such as substituting "copyrighted content" with alternative expressions. This defense strategy effectively prevents the model from retaining copyrighted data in its memory.

Questions:
1. How applicable and universal is the proposed mechanism? MovieTection relies on box office rankings, neglecting niche or independent films, which could lead to an underestimation of long-tail copyright risks. Additionally, the impact of image compression and resolution changes on detection results has not been tested, affecting the robustness of the method in real-world deployment, and preventing assurance of its effectiveness in diverse scenarios.

2. Does it involve ethical issues? Although the publication of 14,000 movie frames is considered "fair use," this approach may still provoke copyright disputes. The article lacks sufficient ethical argumentation and does not fully address potential copyright conflicts or privacy concerns, which may expose involved parties to unnecessary legal risks.

3. Can it effectively prevent potential attacks? The current study only simulates key forgery and fine-tuning attacks, without considering more complex threats such as adversarial samples or model distillation. This results in a potentially insufficient defense capability, unable to comprehensively address various security threats, thus leaving certain security vulnerabilities.

4. What is the computational overhead? The paper does not analyze the computational cost introduced by multiple frame inputs, particularly the impact on GPU memory. This lack of a detailed evaluation could limit the application of the method in low-resource environments, leading to performance bottlenecks in real-world deployment and hindering its widespread use in resource-constrained settings.

5. Is it aligned with specific copyright laws (such as fair use exceptions)? The paper does not integrate specific copyright legal provisions, such as fair use exceptions, leading to ambiguity in the legal significance of its conclusions. The lack of consideration for the broader legal framework, especially regarding fair use and its impact on copyright protection, results in an unclear legal interpretation and a lack of in-depth analysis of the real-world legal implications.

**Relation To Broader Scientific Literature:**

Yes. The issue regarding Relation To Broader Scientific Literature will be elaborated on in the "Questions For Authors" section.

**Theoretical Claims:**

Yes. The issue regarding Theoretical Claims will be elaborated on in the "Questions For Authors" section.

---

> ### Author Rebuttal · Authors · 2025-03-31
>
> Dear Reviewer,
>
> We appreciate the time and effort dedicated to evaluating our paper. We understand the concerns raised and below, we address each point in detail:
>
> > **Q1.1** How applicable and universal is the proposed mechanism?
> >
>
> We acknowledge that MovieTection’s focus on box-office hits may limit its coverage of niche films, which are also subject to copyright and may appear in pretraining datasets.
>
> We would like, nonetheless, to make two clarifications:
>
> - While this is indeed a limitation, it reflects a broader challenge shared by most existing studies in the field. Detecting memorization of rarely seen content is inherently more difficult than detecting memorization of content that appears frequently during training.
> - We agree that it is important to assess how VLMs respond to niche content. To complement our experiments, we conducted an auxiliary study in which we tested DIS-CO on three films with little to no international projection.
>
> | Movie ($X = Box Office) | Gaiola Dourada ($3M) | Canção de Lisboa ($930k) | Leão da Estrela ($500k) |
> | --- | --- | --- | --- |
> | GPT-4o | 13% | 8% | 0% |
>
> It’s interesting to see that with DIS-CO, a smaller movie such as *Gaiola Dourada* is still partially recognized by gpt-4o, suggesting that even limited exposure during training can leave some detectable traces. However, the overall trend is clear: as a film’s popularity and likely exposure in training data decreases, so does the strength of the memorization signal. This reinforces our focus on popular films: while weaker signals may appear for niche works, box-office hits offer stronger, more consistent signals for evaluating memorization.
>
> > **Q1.2** The impact of image compression and resolution changes has not been tested.
> >
>
> We did a small experiment to evaluate how different resolutions affect DIS-CO’s performance.
>
> | GPT-4o | 21 Jump Street | 1917 | A Beautiful Mind | A Star is Born | Aladdin | Avg. |
> | --- | --- | --- | --- | --- | --- | --- |
> | 1126x512 | 68% | 86% | 71% | 80% | 92% | **79.4%** |
> | 563x256 | 58% | 85% | 66% | 77% | 86% | **74.4%** |
> | 282x128 | 57% | 85% | 58% | 64% | 74% | **67.6%** |
>
> As expected, lower resolutions reduce accuracy, since the model has less visual details to work with. That said, DIS-CO remains effective overall: suspect movies are still clearly distinguishable from clean ones, which consistently score near 0% accuracy, regardless of the resolution. Although smaller frames introduce a light performance drop, they can still be a practical choice to reduce computational effort without significantly impacting detection quality.
>
> > **Q2 + Q5** Does it involve ethical issues? Is it aligned with specific copyright laws?
> >
>
> We believe that all aspects of our work are aligned with fair use exceptions and conducted in accordance with relevant ethical and legal standards.
>
> We would like to clarify that the dataset release was reviewed in advance by our institution’s Data Protection Officer (DPO), who provided a positive assessment regarding its compliance.
>
> In addition, to safeguard ethical usage and prevent misuse, the dataset is being released under a restrictive CC BY-NC-SA 4.0 license, which limits its use strictly to non-commercial research and academic purposes.
>
> > **Q3** The current study only simulates key forgery and fine-tuning attacks (…) This results in a potentially insufficient defense capability, thus leaving certain security vulnerabilities.
> >
>
> We respectfully note that our paper does not mention or simulate “key forgery” attacks. Could you clarify what was meant by this term, and how it relates to our work?
>
> Assuming the comment refers to our fine-tuning experiment, we would like to clarify that this component serves primarily as a proof of concept to illustrate that disclosure of memorized content can be mitigated. While we do explore fine-tuning as a way to reduce the disclosure of memorization, this remains a secondary contribution. The central focus of our work is on detection, not mitigation.
>
> > **Q4** The paper does not analyze the computational cost introduced by multiple frame inputs.
> >
>
> Following this suggestion, we evaluated the GPU memory usage when increasing the number of input frames.
>
> | Frames (N) | Qwen2-VL 7B | Qwen2-VL 72B |
> | --- | --- | --- |
> | 1 | 16.00 GB | 138.03 GB |
> | 4 | 17.38 GB | 142.69 GB |
> | **Increase** | **+1.38 GB** | **+4.66 GB** |
>
> On average, each additional frame increases memory by 0.46 GB (Qwen2-VL 7B) and 1.53 GB (Qwen2-VL 72B). This corresponds, for the bigger model, in less than 3.5% total additional memory to process 4 frames, showing that the main memory requirement comes from loading the model itself. The cost added by using multiple frames is small and unlikely to be a problem in most settings.
>
> **Conclusion**
>
> We hope that our responses clarify the concerns raised and demonstrate the validity and value of our work. We thank you for the insightful feedback and we are happy to present further clarifications.

---

> > ### Comment · Reviewer_V24U · 2025-04-03
> >
> > My questions have been clearly answered. Therefore, I change the score to 3. Thanks for your efforts.

---

### Official Review · Reviewer_vDhh · 2025-03-14

**Overall Recommendation:** 4

**Summary:**

This paper introduces DIS-CO, a new method to check if large vision-language models (VLMs) were trained on copyrighted material. the authors use the idea that a model will "remember" specific content if it has seen it before. In this work, the model is asked to name a movie from a single frame or caption. The authors build a new dataset called MovieTection, which includes 14,000 movie frames (and captions) from films released before and after the model’s training cutoff date. Their experiments show that when the model has seen a movie during training, it is more likely to correctly name the movie, and this method works well across different types of models, even in settings where frames are very challenging and we can only see a few of the model’s outputs.

## update after rebuttal
The author clearly addressed my question so I kept the score as accept. I do think this paper is a valuable contribution to the community, and I don't agree with some other reviewers who claim that the paper lacks novelty or is too limited due to movie datasource.

**Claims And Evidence:**

Core Claims:

1. DIS-CO effectively detects the inclusion of copyrighted movies in a model’s training data, outperforming baseline approaches across multiple evaluation metrics.
2. The proposed method is applicable in both white-box and black-box settings.

Evidence:

1. Experiments on the MovieTection benchmark demonstrate that models yield significantly higher accuracy and AUC when queried with suspect frames (from copyrighted movies) than with clean or non-member frames.
2. The study shows a clear improvement in detection performance with longer prompt contexts and reveals a positive correlation between factors such as movie popularity (box-office revenue) and quality (IMDb ratings) with the likelihood of memorization.

**Essential References Not Discussed:**

All essential references I know have been discussed.

**Experimental Designs Or Analyses:**

1. The experimental design is comprehensive, using a newly constructed benchmark (MovieTection) alongside an existing dataset (VL-MIA/Flickr) to evaluate model performance.
2. The study considers various dimensions including prompt context length, movie popularity, and quality, offering insights into factors that influence model memorization.
3. Human evaluation experiments are conducted to differentiate between generalization and genuine memorization, adding depth to the analysis.

**Methods And Evaluation Criteria:**

Methods:

1. The approach involves querying VLMs with free-form text prompts where the model is asked to identify the movie from an input frame or caption.
2. DIS-CO distinguishes between “suspect”  and “clean” movies based on their release dates relative to the model’s training cutoff.
3. Two variants are proposed: one that considers all correct predictions (DIS-CO) and another (⌊DIS-CO⌋) that filters out cases where both image-based and caption-based queries agree, thereby reducing bias.

Evaluation:

1. Performance is assessed using accuracy and Area Under the Curve (AUC) metrics on the MovieTection benchmark and the VL-MIA/Flickr dataset.
2. Experiments span several models (e.g., GPT-4o, Gemini-1.5 Pro, Qwen2-VL 72B, LLaMA-3.2 90B)

Weakness: The paper focuses on movie frames and captions, and it remains to be seen if the approach can generalize to other kinds of copyrighted content or other multimodal tasks.

**Other Comments Or Suggestions:**

no other comments. the paper is well-written.

**Other Strengths And Weaknesses:**

All strengths and weakness has been pointed out above.

**Questions For Authors:**

1. My question to this paper is whether if this paradigm can be applied to other copyright content as well beyond just movie, which also states in the method review section:
    "The paper focuses on movie frames and captions, and it remains to be seen if the approach can generalize to other kinds of copyrighted content or other multimodal tasks."

**Relation To Broader Scientific Literature:**

The work builds on a rich literature on membership inference attacks and data contamination detection, extending these ideas from text-only models to the multimodal setting of VLMs. It relates to recent prompting approaches and entropy-based methods while addressing their limitations, especially in black-box scenarios.

**Theoretical Claims:**

There is no theoretical claims in this paper

---

> ### Author Rebuttal · Authors · 2025-03-31
>
> Dear Reviewer,
>
> We greatly appreciate the time and effort you invested in reviewing our paper. Below, we provide a response to your question.
>
> > My question to this paper is whether if this paradigm can be applied to other copyright content as well beyond just movie, which also states in the method review section: "The paper focuses on movie frames and captions, and it remains to be seen if the approach can generalize to other kinds of copyrighted content or other multimodal tasks.”
> >
>
> We believe it definitely can! In fact, we even ran DIS-CO on a small experiment with a completely different type of visual content: comic books.
>
> The works we used for this were: *Astérix Legionary;* *Lucky Luke: Billy the Kid;* *The Amazing Spider-Man #1;* *Spirou and Fantasio: Comme Zorglub;* and *Tintin in America*.
>
> Given that these comics come from series with multiple of similar-looking volumes, we expected a model to struggle. Still, GPT-4o managed to correctly identify the specific comic-title surprisingly often.
>
> |  | Astérix | Lucky Luke | Spider-Man | Spirou and Fantasio | Tintin |
> | --- | --- | --- | --- | --- | --- |
> | GPT-4o | 52.0% | 61.3% | 42.3% | 68.8% | 67.2% |
>
> That said, we ended up leaving this part out of our main experiments because, unlike movie frames, where the visual signal is (mostly) isolated, comic books introduce an extra variable: text inside the images. This makes it harder to know whether the model’s prediction came from the visual content or if it was leveraging the text. Since our goal was to test visual memorization specifically, we felt that mixing modalities here could weaken the core message.
>
> **Conclusion:**
>
> We hope that our answer has addressed your concern. Please let us know if any further clarification or additional information is needed from our end.

---

### Official Review · Reviewer_tTxp · 2025-03-17

**Overall Recommendation:** 3

**Summary:**

This paper investigates the challenge of verifying whether copyrighted content was used to train large vision-language models (VLMs) without direct access to their training data. The authors introduce DIS-CO, a novel approach that leverages the hypothesis that VLMs can recognize images from their training corpus. By systematically querying a VLM with specific frames from copyrighted material, DIS-CO extracts content identity through free-form text completions. To evaluate its effectiveness, the authors present MovieTection, a benchmark containing 14,000 frames with detailed captions from films released both before and after a model’s training cutoff. Experimental results demonstrate that DIS-CO significantly enhances detection performance, nearly doubling the AUC of the best prior method for models with available logits.

**Claims And Evidence:**

Claim: The paper claims that a VLM is able to recognize images from its training corpus.
This key claim is supported by the evidence in existing literature [r1] that data encountered during training leads to greater model confidence when generating outputs

[r1] Li Z, et al. Membership Inference Attacks against Large Vision-Language Models. NeurIPS, 2024.

**Essential References Not Discussed:**

The paper provides a well-contextualized discussion of prior work and thoroughly cites relevant research in membership inference attacks (MIA), dataset attribution, and multimodal model evaluation.

**Ethical Review Concerns:**

copyright issue of the collected data

**Ethical Review Flag:**

Flag this paper for an ethics review.

**Ethics Expertise Needed:**

["Legal Compliance (e.g., GDPR, copyright, terms of use)"]

**Experimental Designs Or Analyses:**

1.	Validity of Experimental Setup.
Strengths: The experimental design of **DIS-CO** is well-structured to test its effectiveness in detecting copyrighted content within VLMs. The evaluation setup appropriately considers **both white-box and black-box models**, making the results more generalizable.
Weakness: However, a potential limitation is the **assumption that chronological release serves as a strict boundary for training data exposure**. While this is a reasonable heuristic, it does not entirely rule out indirect exposure through publicly available media (e.g., trailers, posters). A complementary **dataset contamination check** would strengthen the claim that post-cutoff movies are truly novel to the model.
2.	Baseline Comparisons.
Strengths: The study effectively benchmarks **DIS-CO** against prior methods, including **MCQA and Renyi entropy-based techniques**. The **AUC and accuracy metrics** provide a fair and interpretable comparison. The decision to remove **caption-based correct predictions** from the evaluation is a useful refinement, ensuring that memorization is attributed to visual data rather than textual association.
Weakness: An area for improvement is the **lack of an ablation study** that isolates the contributions of **prompting strategy, query selection, and frame type**. Understanding the relative impact of these components would clarify why DIS-CO outperforms prior approaches.

**Methods And Evaluation Criteria:**

The proposed methods and evaluation criteria are well-aligned with the problem statement. The study introduces a rigorous benchmark, strong comparative baselines, and a method applicable to real-world black-box models. Improvements in ablation studies, dataset validation, and robustness against adversarial probing would further enhance its impact.

1.	Appropriateness of the Proposed Method (DIS-CO). The DIS-CO method is well-suited for detecting whether large vision-language models (VLMs) have been trained on copyrighted content. By leveraging free-form text generation, the approach mitigates biases introduced by multiple-choice settings and allows for more natural model responses.
2.	Strength of Benchmark Dataset (MovieTection). The introduction of MovieTection as a benchmark dataset is a strong contribution. It is carefully constructed with a temporal split to differentiate between pre-training and post-training data. The inclusion of "main frames" and "neutral frames" enhances evaluation granularity, making the benchmark robust for detecting training exposure.
3.	Evaluation Metrics and Comparisons. The study effectively evaluates DIS-CO using AUC scores, accuracy metrics, and comparative baselines (MCQA, Rényi method, Captions-based prompting). The removal of caption-based correct predictions to isolate image memorization is a novel refinement.

**Other Comments Or Suggestions:**

The paper is well-organized and easy to follow. The sections are logically structured, guiding the reader through the problem statement, proposed method, experiments, and results.

**Other Strengths And Weaknesses:**

Strengths:
1) The MovieTection dataset is a new benchmark designed for testing model memorization of copyrighted materials, addressing a crucial gap in dataset attribution research.
2) DIS-CO achieves state-of-the-art performance in detecting memorization, improving upon prior methods such as MCQA and Rényi entropy-based inference.
3)The methodology is clearly explained, with detailed steps for dataset construction, query formulation, and evaluation.


Weakness:
1) While DIS-CO’s empirical results are strong, the paper lacks formal mathematical analysis of why free-form text completion is a more reliable indicator of memorization than multiple-choice formats.
2)Additional theoretical justification (e.g., probability bounds on model memorization) could strengthen the claims.
3) The study does not dissect the impact of different components of DIS-CO, such as the number of frames per query, prompt variations.

**Questions For Authors:**

1.	Clarification on Dataset Contamination.The paper assumes that movies released after the training cutoff date are not included in the model’s training data. However, some models might have encountered promotional content (e.g., trailers or posters) that could influence performance on recent movies. Can you provide more details or experiments to assess the potential contamination from such external content sources?
2.	Ablation Study for DIS-CO. You have provided solid results showing DIS-CO’s advantage over prior methods. However, could you consider conducting an ablation study to isolate the contributions of specific components of DIS-CO (e.g., the number of frames per query, prompt variations)?

**Relation To Broader Scientific Literature:**

DIS-CO builds on membership inference attacks as explored by [r1] and [r2], extending them to black-box models and offering more reliable detection of copyrighted content in VLMs, surpassing previous methods like Rényi entropy.

[r1] Li Z, et al. Membership Inference Attacks against Large Vision-Language Models. NeurIPS, 2024.
[r2] Pinto F, et al. Extracting Training Data From Document-Based VQA Models. ICML, 2024.

**Theoretical Claims:**

1. Soundness of the DIS-CO Method. The paper’s central theoretical claim—that free-form text completion provides a stronger signal for detecting memorized content than multiple-choice settings—is intuitively reasonable and supported by past studies on language model leakage. However, The paper lacks a formal proof of why free-form completions reduce false positives better than MCQA.
2. Upper-Bound Estimation of Memorization. The approach of defining DIS-CO and ⌊DIS-CO⌋ as lower and upper bounds of memorization is reasonable, but the paper does not theoretically prove the validity of this bounding technique.

---

> ### Author Rebuttal · Authors · 2025-03-31
>
> Dear Reviewer,
>
> Thank you very much for your valuable feedback and comments. Below, we address each of your questions.
>
> > **W1.** Lack of formal mathematical analysis.
> >
>
> Here, we present our mathematical analysis to support our intuition on why free-form completions (FF) are a more robust indicator of memorization than Multiple Choice (MC) formats.
>
> **Random-Chance Baselines:**
> In an MC setting with $k$ options, the probability of a correct guess by chance is $P_{MC} = \frac{1}{k}$, wich for $k=4$ would result in 25%
>
> By contrast, in an FF setting the model must generate the exact label from a vast output space $\Omega$ (with $|\Omega| \gg k$), so that the chance-level probability is approximately $P_{FF} \approx \frac{1}{|\Omega|},$ which is orders of magnitude lower than $P_{MC}$.
>
> **Impact of Non-Uniform Priors:**
> It is true that models may exhibit a bias toward more popular movies. Even if this bias increases the chance for a popular movie by a factor of 100, the overall probability remains extremely small. For instance, if $|\Omega| = 10{,}000$, even with a bias, the probability $P'_{FF} = \frac{100}{10{,}000} =$  1%, which is still 25 times lower than the MC chance of 25%.
>
> > **Q1.** Clarification on Dataset Contamination.
> >
>
> We fully agree that external sources could influence the model’s performance. To investigate this potential contamination, we would like to draw your attention to Appendix D. We found that GPT-4o (with a knowledge cutoff in October 2023) acknowledged 20 out of the 50 clean movies included in MovieTection.
>
> We believe it's reasonable to assume these 20 titles are likely candidates for having appeared in publicly accessible sources such as trailers, promotional posters, or press coverage before the model's cutoff date, though not as full movies, since they had not been released yet.
>
> The table below summarizes findings from Table 10 in Appendix D.
>
> | Release Period | # of Movies | Non-zero Values | Max Accuracy |
> | --- | --- | --- | --- |
> | **Nov 2023 - Feb 2024** | 6 | 4 | 6% |
> | **Mar–Oct 2024** | 12 | 0 | 0% |
>
> We observe that movies released closer to the model's cutoff date, and thus with more external content available, result on the highest accuracy scores, while very recent releases, for which public information was considerably sparser at the cutoff date, show no correct mappings. The highest observed accuracy for this subset of recognized movies was, however, only 6%, significantly lower than accuracy scores typically achieved for suspect movies. Therefore, we firmly believe that exposure to external content alone minimally impacts our primary results.
>
> > **W2 + Q2** Could you consider conducting study to isolate the contributions of specific components of DIS-CO?
> >
>
> Isolating and understanding the contributions of individual components in DIS-CO is definitely essential. However, we believe to have already explored some of these aspects, for which we would like to direct your attention to Appendix I, where we analyze the effect of varying the number of frames per query. Our analysis is conducted across the different frame types (main vs. neutral) and model sizes, offering a wide view of how these variables affect performance.
>
> As for your suggestion regarding prompt variations, we conducted an additional experiment to evaluate model sensitivity to different wordings, which can be relevant for practical applications, since user inputs will naturally have some variance in the words.
>
> We designed a small-scale evaluation using two categories of prompts:
>
> - Biased Prompts: These include additional cues that might assist the model.
>     - Example: “What Oscar-winning movie is this frame from?”
> - Paraphrased Prompts: Semantically equivalent rephrasings of the default prompt.
>     - Example: “Can you identify what movie is present here?”
>
> We evaluated model responses on a subset of the MovieTection dataset, summarized below:
>
> | Prompt Type | 21 Jump Street | 1917 | A Beautiful Mind | A Star is Born | Aladdin | Avg. |
> | --- | --- | --- | --- | --- | --- | --- |
> | Easier | 83% | 100% | 87% | 85% | 92% | 89.4% |
> | Default | 68% | 86% | 71% | 80% | 92% | 79.4% |
> | Default Paraphrased | 60% | 88% | 74% | 82% | 92% | 79.2% |
>
> Given that in real-world settings the target content may not always be a well-known blockbuster, biasing the model toward popular titles through hints may not be ideal. We believe that sticking to neutral prompt variations is a more reliable choice, as it avoids introducing external priors and better reflects the model’s actual memorization.
>
> > **Flag for Ethics Review**
> >
>
> Regarding the ethics review flag, please refer to our response provided in the reply to Reviewer V24U (Q2 + Q5), which addresses this issue as well.
>
>
> **Conclusion:**
>
> We hope that our answers have addressed your concerns, and thank you once again for your valuable feedback.
>
> Please let us know if any further clarification or additional information is needed from our end.

---

### Decision · Program_Chairs · 2025-05-01

**Decision:**

Accept (poster)

**Comment:**

This paper proposes DIS-CO, a method for detecting whether copyrighted content was used in VLM training via free-form prompting, and introduces the MovieTection benchmark.

The majority of reviewers (4, 3, 3, 1) found the paper well-executed, with strong results and clear value. While one reviewer raised concerns about novelty and framing, the authors provided rebuttals and showed some generalization beyond movies. Ethics reviewers flagged minor concerns around dataset usage, which the authors have committed to address in the final version.

Overall, the ACs recommend acceptance, contingent on revision to address the ethical concerns raised (e.g., clearer dataset documentation and legal framing) as promised in the rebuttal.